# PoET: A generative model of protein families as sequences-of-sequences

**Timothy F. Truong Jr**
OpenProtein.AI
NY, USA
ttruong@openprotein.ai

**Tristan Bepler**
OpenProtein.AI
NY, USA
tbepler@openprotein.ai

## Abstract

Generative protein language models are a natural way to design new proteins with desired functions. However, current models are either difficult to direct to produce a protein from a specific family of interest, or must be trained on a large multiple sequence alignment (MSA) from the specific family of interest, making them unable to benefit from transfer learning across families. To address this, we propose **Pro**tein **E**volutionary **T**ransformer (PoET), an autoregressive generative model of whole protein families that learns to generate sets of related proteins as sequences-of-sequences across tens of millions of natural protein sequence clusters. PoET can be used as a retrieval-augmented language model to generate and score arbitrary modifications conditioned on any protein family of interest, and can extrapolate from short context lengths to generalize well even for small families. This is enabled by a unique Transformer layer; we model tokens sequentially within sequences while attending between sequences order invariantly, allowing PoET to scale to context lengths beyond those used during training. In extensive experiments on deep mutational scanning datasets, we show that PoET outperforms existing protein language models and evolutionary sequence models for variant function prediction across proteins of all MSA depths. We also demonstrate PoET's ability to controllably generate new protein sequences. [1]

## 1  Introduction

Proteins carry out most of the biological functions at the molecular level of life. The function of a protein is encoded by its specific amino acid sequence and the three-dimensional structure that the sequence folds into. Engineering proteins for novel and enhanced function is a key problem in pharmaceuticals and biotechnology and involves designing novel sequences or modifying existing natural proteins for these purposes. Deep mutational scans and directed evolution experiments have been used to successfully design novel proteins [1–3], but can be costly and difficult to implement, which makes these experimental methods inapplicable for many proteins and functions of interest. Accurate computational models of sequence-function relationships can narrow down the protein sequence search space, reduce the need for expensive experiments, and enable the design of more novel proteins and functions.

Protein language models have emerged as promising methods for understanding and designing protein sequences [4–7]. In particular, generative models offer a natural way to produce new protein designs. By training on large corpuses of natural protein sequences, these models learn evolutionary constraints on sequence space. They can then be used either to generate realistic sequences directly by sampling [8, 9], or to identify promising protein sequence variants by predicting the relative fitness of the variants of interest using the sequence likelihoods as a proxy [4, 10, 11].

---

[1]Code and pre-trained model weights are available at https://github.com/OpenProteinAI/PoET.

37th Conference on Neural Information Processing Systems (NeurIPS 2023).

Traditionally, family-specific models learn evolutionary constraints specific to the protein family of interest by training on a multiple sequence alignment (MSA) of homologous sequences [12, 13]. However, this is ineffective for protein families with few sequences due to the lack of sufficient training data and inability to exploit information across families. These models also assume that MSAs are accurate, and cannot model novel insertions or deletions (indels) not present in the training MSA. More recently, unconditional large protein language models [5, 10, 11] have been developed. Trained on all known natural protein sequences, unconditional protein language models generalize across protein families. However, these unconditional single-sequence models cannot be easily directed to generate a protein from a specific family of interest, and underperform family-specific models for relative fitness prediction. Hybrid models such as Tranception [11] and TranceptEVE [14] combine unconditional language models with family-specific models to enable specialization to protein families. Nonetheless, it is unclear how to use these models to generate sequences with novel indels, and predictions from the family-specific models do not directly benefit from transfer learning across families.

Here, we propose the **Pro**tein **E**volutionary **T**ransformer (PoET), an autoregressive generative model of whole protein families that addresses these limitations. By learning to generate sets of related proteins as sequences-of-sequences across tens of millions of natural protein sequence clusters, PoET is able to generalize about evolutionary processes *across* protein families, and avoids issues related to conditioning on MSAs. In order to capture conditioning between sequences in an order independent manner (the order of sequences within a family is arbitrary) and to generalize to large context lengths, we propose a novel Transformer layer (§3.1.2) that models order-dependence between tokens within sequences and order-independence between sequences. PoET has the following properties:

- PoET can be used as a retrieval-augmented protein language model by conditioning the model on sequences from any family of interest. This also allows PoET to be used with any sequence database and to incorporate new sequence information without retraining.
- PoET is a fully autoregressive generative model, able to generate and score novel indels in addition to substitutions, and does not depend on MSAs of the input family, removing problems caused by long insertions, gappy regions, and alignment errors.
- By learning across protein families, PoET is able to extrapolate from short context lengths allowing it to generalize well even for small protein families.
- PoET can be sampled from and can be used to calculate the likelihood of any sequence efficiently.

We demonstrate these properties and show that PoET outperforms existing protein language models and evolutionary sequence models for variant effect prediction in extensive experiments on the 94 deep mutational scanning datasets in ProteinGym. PoET improves substitution effect prediction across proteins of all MSA depths, and also improves effect prediction of indels. A simple weighted average ensemble of PoET with existing methods further improves performance both across MSA depths and in sequences with large numbers of mutations. Furthermore, when used for generation, PoET controllably produces diverse, novel sequences that more closely match natural statistical constraints, resulting in better folding pLDDTs, than other generative protein language models. We expect PoET to become integral to future protein mutagenesis efforts.

## 2 Related Work

**Evolutionary sequence models** are well established methods in biological sequences analysis. To model protein families, these models search large protein sequence databases for homologs, align the positions of these homologs in an MSA, and then fit statistical sequence models to the MSA. Common models include site independent models [15], profile HMMs [16], and coupling models [17]. Newer variants incorporate higher order correlations between sequence positions by training a VAE [13] or by building phylogentic trees [12]. These approaches are often referred to as "alignment-based" and must be fit on a family-by-family basis, requiring large numbers of members to generalize. A significant limitation of these models is that they assume the MSA is an accurate model of the evolutionary process generating the sequences, when in fact, MSA algorithms inevitably make alignment errors; regions with long insertions or lots of gaps can be particularly problematic. Furthermore, they cannot model novel insertions or deletions (indels) that are not present in the training MSA.

**Unconditional protein language models** that do not condition on homologs at inference have emerged as powerful methods for understanding and generating protein sequences. Both bidirectional models [4, 6, 18–20] and autoregressive generative models [5, 10, 11] have demonstrated competitive performance for variant function prediction. The latter type of model has the advantage of being able to score indels, but both cannot integrate evolutionary context not present in the trained model parameters. In contrast to family-specific evolutionary sequence models trained on sequences derived from a specific protein family [13, 17, 21], these models are trained on large protein databases [22, 23] that span all known sequences. This enables them to learn evolutionary constraints that generalize across families to improve predictions for small families with few homologs, but they generally underperform family-specific models for larger families.

**Conditional protein language models** fit between the unconditional and family-specific paradigms. Only a few works have explored this direction to date. Masked language models of whole MSAs are able to integrate evolutionary context directly for conditioning site predictions [24], but are unable to model insertions in the alignment. Ram and Bepler [25] use an encoder-decoder framework to generate new sequences conditioned on an MSA, which removes the insertion limitation of Rao et al. [24], but still requires conditioning on aligned input sequences. Notin et al. [11] combine predictions from an unconditional language model and an alignment-based model and show that integrating retrieval-based methods with protein language models can improve variant function prediction performance. However, the reliance on an alignment-based model means that the combined model is still limited by the constraints of MSAs.

**Retrieval-augmented language models** have shown impressive results in natural language processing, especially on Question Answering tasks. These models incorporate a database search as part of the text generation process in order to generate new text conditioned on prototypes found in the database [26–32]. In this way, they are conceptually similar to the conditional protein language models above. Retrieval-augmented approaches have the advantage of not requiring the entire training corpus to be encoded within the model parameters and the ability to easily integrate new data without retraining by simply adding it to the retrieval database. We adopt a similar approach with PoET in order to realize similar benefits. However, we build on well-established, fast, and accurate protein search tools for retrieval and frame the protein sequence generation problem as a sequence-of-sequences problem to incorporate retrieved-sequence conditioning, a fundamentally different paradigm than that employed by current retrieval-augmented models in natural language processing.

## 3 The Protein Evolutionary Transformer (PoET)

PoET is an autoregressive generative model of the distribution over protein families, where each family is generated as a sequence-of-sequences. Specifically, it models the distribution $P(X = x)$, where $x = s_1, s_2, ..., s_n$ is the concatenation of $n$ sequences $s_i$ from the same family, and each sequence $s_i = s_{i,1}, s_{i,2}, ..., s_{i,L_i}$ is a sequence of $L_i$ amino acids padded by a start and end token. For example, below is a sequence of three protein sequences of lengths $4$, $6$, and $5$ with start token denoted by $ and stop token denoted by $*$:

| $ | M | K | * | $ | M | H | I | P | * | $ | M | P | V | * |
|---|---|---|---|---|---|---|---|---|---|---|---|---|---|---|
| $s_{11}$ | $s_{12}$ | $s_{13}$ | $s_{14}$ | $s_{21}$ | $s_{22}$ | $s_{23}$ | $s_{24}$ | $s_{25}$ | $s_{26}$ | $s_{31}$ | $s_{32}$ | $s_{33}$ | $s_{34}$ | $s_{35}$ |

$$\underbrace{\qquad}_{s_1} \qquad \underbrace{\qquad}_{s_2} \qquad \underbrace{\qquad}_{s_3}$$

When referring to a sequence-of-sequences, $s_i$, which has one index $i$, refers to a sequence of tokens - the $i$th sequence in the sequence-of-sequences, whereas $s_{i,j}$, which has two indices $i, j$, refers to one token, the $j$th token of the $i$th sequence. We use $x$ to denote the full sequence-of-sequences.

PoET generates each token in a sequence $x$ one at a time, decomposing the probability of $x$ as

$$P(x) = P(s_1, s_2, ..., s_n) = \prod_{i=1}^{n} P(s_i|s_{<i}) = \prod_{i=1}^{n}\prod_{j=1}^{L_i} P(s_{i,j}|s_{<i}, s_{i,<j}) \tag{1}$$

The order of the individual sequences in a sequence-of-sequences is arbitrary, and we propose a novel Transformer-based architecture to exploit this order invariance.

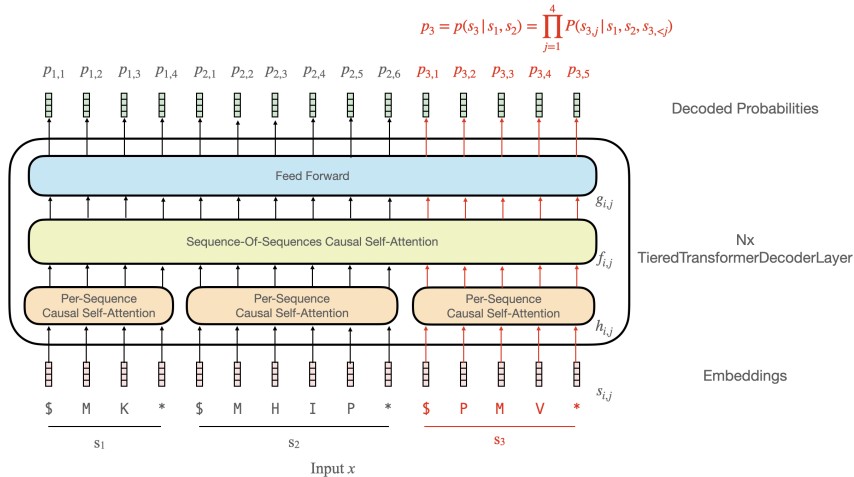

(a) PoET generates sets of homologous proteins as sequences-of-sequences. The equation in red demonstrates how to compute the conditional probability of a sequence given homologous sequences.

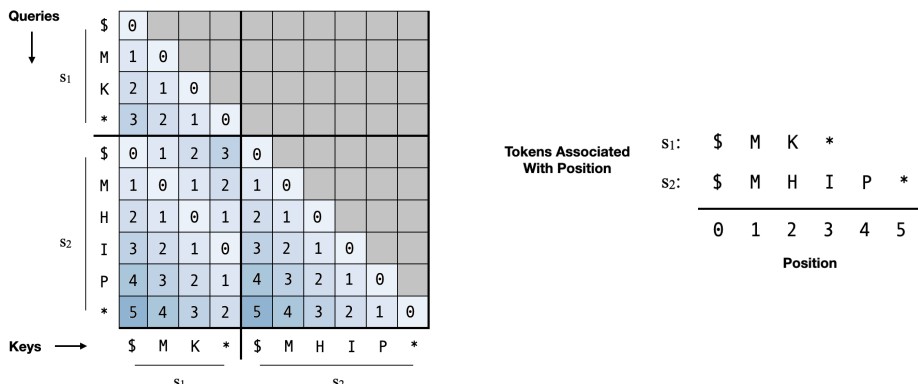

(b) Inter-sequence relative position encoding scheme used in sequence-of-sequences self-attention that is invariant to sequence ordering. (*Left*) Relative positions between tokens in a sequence-of-sequences composed of two sequences. (*Right*) Tokens associated with absolute positions.

Figure 1: PoET Architecture

## 3.1 Model Architecture

We propose a variant of the common Transformer decoder layer (Figure 1a) to capture order invariance between sequences while preservering order-dependence between tokens within sequences. We accomplish this using two attention modules: (1) a within-sequence module in which the representation at each position of each sequence is updated based on attending only to the other tokens within this sequence, and (2) a between-sequence module in which the representation at each position of each sequence is updated based on attending to other sequences within the sequence-of-sequences (§3.1.2). This tiered approach is critical for capturing long-range dependencies between sequences and uniquely allows our model to extrapolate to much longer context lengths than used during training (§5.2.2), improving sequence generation and performance on downstream tasks (Appendix H). Our full PoET model is a stack of these layers with causal self-attention.

### 3.1.1 Input Embedding

The input sequence $x = s_{i,j}, i \in 1..n, j \in 1..L_i$ of amino acids and start/stop tokens is first converted into a sequence of continuous embeddings $h_{i,j}$ by mapping each token to its learned embedding:

$$h_{i,j} = W_{s_{i,j}}, s_{i,j} \in \text{AA} \cup \{\text{START}, \text{STOP}\} \tag{2}$$

$$W \in \mathbb{R}^{|\text{AA} \cup \{\text{START}, \text{STOP}\}| \times d}$$

where `AA` is the set of 20 standard amino acids, and $W$ is a matrix of learnable embeddings of dimension $d$.

### 3.1.2 Tiered Transformer Decoder Layers

Next, the embeddings $h_{i,j}$ are transformed by $N$ layers of `TieredTransformerDecoderLayers` (Appendix Algorithm 1), a novel layer for processing a sequence-of-sequences that is invariant to the order of the individual sequences, and extrapolates to context lengths substantially longer than the training context length.

The `TieredTransformerDecoderLayer` is composed of two phases. In the first phase, causal self-attention is applied independently to each sequence $h_i$ of the input sequence-of-sequences, transforming them into new sequences $f_i = \text{PerSequenceSelfAttn}(h_i)$. Relative positional information is encoded by applying Rotary Positional Encodings (RoPE) [33] to the queries and keys before applying self-attention in the standard manner; the absolute position for $f_{i,j}$ is $j$.

The second phase applies causal self-attention to the entire sequence-of-sequences by concatenating the individual $f_i$ from the previous layer into one sequence before applying self-attention: $g_{i,j} = \text{SequenceOfSequencesSelfAttn}([f_{<i}; f_{i,<j}])$. In order to make self-attention in this phase invariant to sequence order, we adopt a simple but novel inter-sequence relative positional encoding scheme (Figure 1b): for $g_{i,j}$ the absolute position is $j$. Just as in the first phase, the absolute position for tokens in the $i$th sequence $g_i$ is independent of the position $i$ of the sequence in the sequence-of-sequences. Thus, the positional information encoded by RoPE in this layer alone *does not distinguish* between the positions of tokens in *different* sequences. For example, the relative position between the first token of the first sequence $f_{1,1}$ and the first token of the second sequence $f_{2,1}$ is 0. The fact that these two tokens come from two different sequences is encoded by the first phase, which operates on the two sequences independently. This inter-sequence relative positional encoding scheme has two useful properties:

1. it encodes the fact that amino acids at similar absolute positions in homologous proteins are more likely to be drawn from the same distribution

2. it limits the maximum relative position encoding needed to the number of tokens in an *individual* protein sequence[2], rather than the number of tokens in a sequence-of-sequences, allowing the model to generalize to longer sequences-of-sequences than seen during training

### 3.1.3 Decoded Probabilities

Lastly, the output from the last `TieredTransformerDecoderLayer`, $g_{i,j}$, is decoded into token probabilities by applying a linear transformation $P(s_{i,j}|s_{<i}, s_{i,<j}) = p_{i,j}(s_{i,j}) = \text{Linear}(g_{i,j})$. Here, $p_{i,j}$ is a vector of probabilities, one for each distinct token $\in \text{AA} \cup \{\text{START}, \text{STOP}\}$, and $p_{i,j}(s_{i,j})$ is the probability of the token $s_{i,j}$ according to $p_{i,j}$.

### 3.2 Training Data

Models were trained on 29 million sets of homologous sequences. Each set corresponds to a sequence in UniRef50 Version 2103, and contains all its homologs in UniRef50 found using Diamond [34]. We removed any sets with fewer than 10 homologs. To avoid overfitting on promiscuous sequences which may belong to a large number of sets, we sample each set with weight inversely proportional to the size of the set ("inverse count" sequence weighting). The order of sequences within a sequence-of-sequences is randomized to promote order invariance. See Appendix B for more details.

## 4 Protein Variant Fitness Prediction

Protein variant fitness prediction is the task of assigning a score to each sequence in a set of variants $\{v_1, v_2, ..., v_n\}$ of a target sequence $t$ that accurately reflects the relative fitness of the variants. A protein variant $v_i$ can be any sequence with a limited number of substitutions, insertions, and/or deletions relative to the target $t$ that the experimenter believes may have improved fitness. Fitness

---

[2]While the length of a protein is unbounded (but finite), in practice, the vast majority ($> 99\%$) of proteins are less than 1024 amino acids long [11].

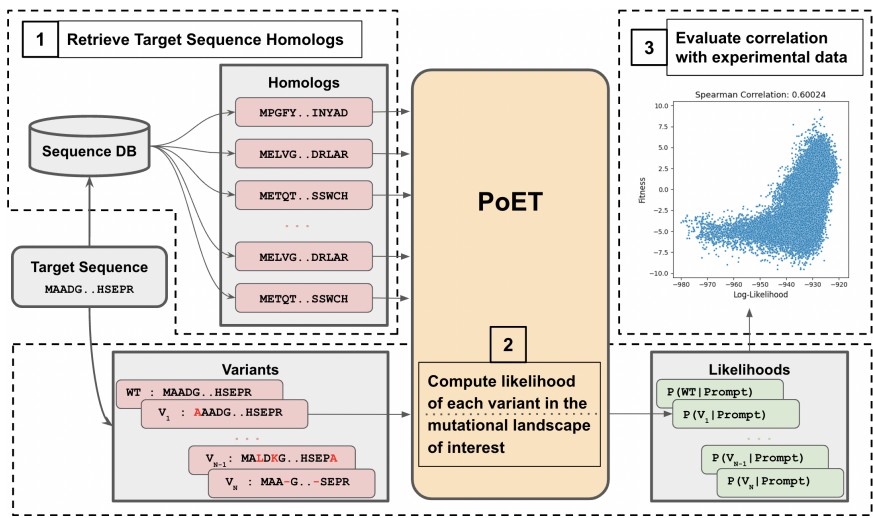

Figure 2: Illustration of evaluating PoET for variant fitness prediction on a DMS dataset

refers to the value of any property of a protein sequence related to function that the experimenter is interested in optimizing e.g. thermostability, enzymatic activity, etc.

**Benchmarking using deep mutational scans** Deep mutational scanning (DMS) is a method for measuring the fitness properties of thousands to millions of variants of a protein using next-generation sequencing techniques [1]. Data from these experiments have been used to evaluate fitness prediction methods. We evalaute PoET on ProteinGym [11], the largest collection of such data yet, containing 87 datasets with substitution variants and 7 datasets with indel variants. We use the same validation set as Notin et al. [11] for tuning hyperparameters.

**Fitness Prediction with PoET** PoET predicts the fitness of a variant as the conditional log-likelihood of the variant $v_i$ given a set of sequences $S$ homologous to the target $t$:

$$\hat{F}_i(S = \{s_1, ..., s_m\}) = \log P(v_i|s_1, s_2, ..., s_m) = \sum_{j=1}^{L_i} \log P(v_{i,j}|s_1, s_2, ..., s_m, v_{i,<j}) \quad (3)$$

The set $S$ is retrieved by searching a large database of proteins such as UniRef100 [23] for sequences homologous to $t$. The homologs form a diverse set of sequences that define the protein family of interest. By conditioning on these sequences, PoET can infer evolutionary constraints on the protein family to improve fitness prediction. The sequences are conditioned on in an arbitrary order. Figure 2 illustrates the evaluation of PoET for variant fitness prediction on one DMS dataset.

Based on validation set performance, we selected the best method for (1) retrieving homologous sequences (Appendix C), (2) subsampling and filtering the homologous sequences to a reasonable context length for efficient inference (Appendix D), and (3) ensembling conditional log-likelihoods computed from different subsamples of homologous sequences. We present results with the best settings in the main results (§5). Our final approach uses the ColabFold protocol [35] for retrieving homologs, and the final fitness prediction score is obtained by averaging the conditional log-likelihoods across subsamples of the full set of retrieved homologous sequences:

$$\hat{F}_{\text{ensemble},i}(S) = \frac{1}{N_{\text{ensemble}}} \sum_{j=1}^{N_{\text{ensemble}}} \hat{F}_i(S_j \subset S) \quad (4)$$

The subsets $S_j$ are drawn by sampling with sequence weights [17], maximum sequence identity to the target $t$ in $\{1.0, 0.95, 0.90, 0.70, 0.50\}$, and total tokens (i.e. context length) in $\{6144, 12288, 24576\}$. All combinations of these parameters are used, for a total of $N_{\text{ensemble}} = 15$.

Table 1: **Average Spearman correlation between model scores and experimental measurements on ProteinGym by MSA depth, and # of parameters in language models.** MSA depth measures the amount of sequence information the MSA of the homologous sequences defining a protein family contains about the target protein (Appendix E). N/A indicates not applicable, whereas a dash (-) indicates applicable, but not computed.

| Model Type | Model name | # Param | Substitutions by MSA Depth | | | | Indels |
| --- | --- | --- | --- | --- | --- | --- | --- |
| | | | Low | Medium | High | All | |
| Align-ment-based | Site independent | N/A | 0.417 | 0.404 | 0.411 | 0.408 | N/A |
| | GEMME | N/A | 0.445 | 0.449 | 0.522 | 0.463 | N/A |
| | EVE (ensemble) | N/A | 0.414 | 0.441 | 0.498 | 0.448 | N/A |
| Uncond-itional PLM | ESM-1v (ensemble) | 3.25B | 0.356 | 0.372 | 0.510 | 0.398 | N/A |
| | ProGen2 (ensemble) | 10.8B | 0.357 | 0.416 | 0.448 | 0.411 | 0.407 |
| | Tranception L (no retrieval) | 700M | 0.377 | 0.399 | 0.429 | 0.401 | 0.430 |
| Cond-itional | MSA Transformer (ens.) | 100M | 0.372 | 0.421 | 0.477 | 0.423 | N/A |
| | PoET (ensemble) | 201M | **0.476** | **0.466** | **0.542** | **0.484** | **0.510** |
| Hybrid | Tranception L | 700M | 0.441 | 0.437 | 0.472 | 0.445 | 0.464 |
| | TranceptEVE M | 300M | - | - | - | - | 0.516 |
| | TranceptEVE L | 700M | 0.454 | 0.463 | 0.508 | 0.471 | 0.466 |
| | PoET (ensemble) + TranceptEVE L | 901M | **0.479** | **0.480** | **0.537** | **0.492** | **0.521** |

## 5   PoET achieves state-of-the-art performance in variant fitness prediction

We evaluate PoET's ability to predict the relative fitness of protein variants by measuring the average Spearman correlation $\bar{\rho}$ between the predicted fitness and the measured fitness across the DMS datasets in ProteinGym. We compare PoET to the best existing alignment-based models, unconditional and conditional protein language models, and hybrid models that combine multiple approaches (Results Summary: Table 1, Per Dataset: Figures S1, S2 ). A brief overview of the baseline methods is provided in Appendix F. Statistical significance was assessed using a paired t-test. For a fair comparison, we ran the relevant baselines with all homologous sequence retrieval methods considered for use with PoET (Appendix C), and present the results for each baseline method paired with its best retrieval method. We do not tune parameters for ensembling the same method with different subsamples of the set of retrieved homologs (Equation 4) because these parameters need to be considered and defined on a per method basis, and many baselines already have their own ensembling strategy as defined by their authors [4, 12, 14].

### 5.1   Comparison to baselines

**Substitutions Benchmark** On the substitutions benchmark, PoET performs comparably to TranceptEVE L, the best performing baseline method (Table 1). PoET improves average Spearman correlation $\Delta\bar{\rho} = 0.013$, but the difference falls just short of statistical significance $p = 0.05029$. Although the methods perform similarly, PoET offers a number of advantages. Inference with PoET is substantially faster when considering on the order of tens to hundreds of thousands of variants (Appendix J), which allows users to more quickly accomplish common tasks such as computing the fitness of all single site substitutions. By avoiding the costly step of training a new model for each protein family, users can more easily experiment with different ways of defining the protein family. For example, users can retrieve the homologous sequences using different sequence search programs and settings, and prior knowledge can be incorporated by selecting the subset of most relevant homologous sequences e.g. sequences from the taxon of interest, or that are known to be functional (Appendix K). Since PoET is able to generalize across protein families, whereas the EVE component of TranceptEVE L cannot, it is less critical that the protein family of interest is defined by a large number of homologous sequences (Appendix C).

On the other hand, if one is willing to spend more time on inference and the protein family is well defined *a priori*, better fitness predictions can be obtained by ensembling PoET with TranceptEVE L (Results: Table 1, Methods: Appendix G). An ensemble of PoET with TranceptEVE L performs significantly better than TranceptEVE L alone ($\Delta\bar{\rho} = 0.021, p < 7\text{e-}6$). The performance improvement is consistent across subsets of the substitutions benchmark broken down by MSA depth (Table 1), mutation depth (Appendix Table S2), and taxon (Appendix Table S3).

We also considered ensembles of PoET with other baselines methods (Appendix Table 7). The ensemble of PoET with GEMME is also notable as it performs similarly to the ensemble with TranceptEVE L on the substitutions benchmark, and requires very little additional time to compute; GEMME is thousands of times faster than either PoET or TranceptEVE L [12]. The main disadvantage is that GEMME is unable to score indels.

**Indels Benchmark** On the indels benchmark, the best performing baseline model is TranceptEVE M, a variation of TranceptEVE L that uses a protein language model with fewer parameters ($\Delta\bar{\rho} = 0.05$). PoET and the ensemble of PoET with TranceptEVE L both outperform TranceptEVE L ($\Delta\bar{\rho} \in [0.044, 0.055]$), and perform comparably to TranceptEVE M ($\Delta\bar{\rho} \in [-0.006, 0.005]$). However, no differences between the aforementioned models are statistically significant. One advantage of PoET is that it is able to not only score indels, but also *generate* sequences with indels (§6). It is also able to score and generate insertions not present in the MSA.

## 5.2 Characterizing the PoET architecture

### 5.2.1 Training Distribution and Model Size

We train variations of PoET with different training distributions and model sizes and investigate their effect on fitness prediction (Figure 3).

**Context Length** We trained 57M parameter versions of PoET for up to 3 days on 7 x A100 GPUs with three context lengths: 4K, 8K, and 16K. Notably, these context lengths far exceed the 1K or 2K context lengths typically used to train other protein and natural language models [5, 11, 24, 36, 37]. We reason that longer context lengths are needed for the model to observe a sufficient number of homologs; even with a context length of 8K, for a sequence of length 500, which is the length of the typical human protein in ProteinGym, the model would only observe ~16 of possibly thousands of homologs. Interestingly, while a context length of 8K performs better than a context length of 4K, increasing the context length further to 16K had a neutral or negative effect on variant effect prediction. It does improve perplexity of heldout sequences.

**Sequence Set Weighting** Next, we trained an 8K context length model using the naive strategy of sampling UniRef50 sequence sets uniformly instead of using our "inverse count" strategy that weights sequence sets based on their size (§3.2). We find that the "inverse count" sampling strategy substantially outperforms the naive strategy. Other methods that also train on sequence sets like MSA Transformer [24] may also benefit from this sampling strategy.

**Model Size** Lastly, using the optimal parameters found thus far, we trained 57M, 201M, and 604M parameter models to explore the effect of model size. Models were trained until validation performance plateaued. Increasing the model size to 201M parameters signficantly improves performance, but increasing the size further to 604M appears to have a neutral effect. We chose the 201M version as the standard PoET model.

### 5.2.2 Model Architecture

To determine the importance of using `TieredTransformerDecoderLayers` (§3.1.2) instead of regular Transformer layers, we trained, on the same task, a regular 201M parameter RoPE-based Transformer that ignores the sequence-of-sequences structure of the input. We compared the generative capabilities of PoET and the Transformer by measuring the perplexity at context lengths both within and beyond the training the context length (Figure 4).

We trained both models with a context length of 8K for 500K steps. As expected based on previous studies[38], the RoPE-based Transformer does not generalize to context lengths beyond the training context length. In contrast, PoET generalizes to ~8x the training context length (61K), which is the longest we tested. PoET also performs better on context lengths within the training context length,

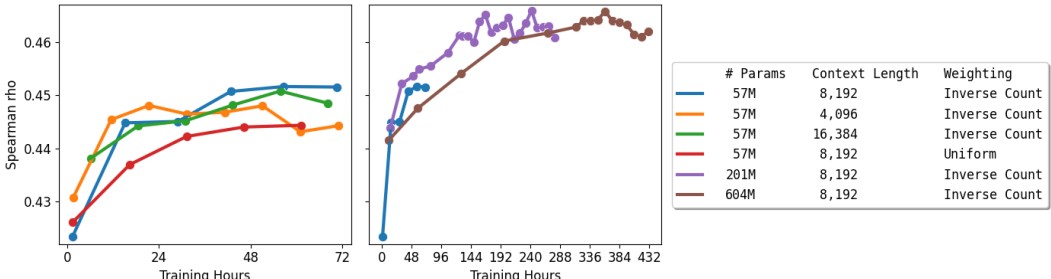

Figure 3: Performance of PoET on the ProteinGym validation set when trained with (*left*) various training distributions and (*right*) model sizes.

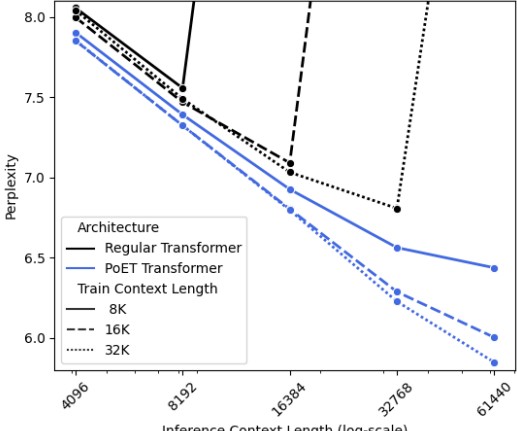

Figure 4: Comparison of the perplexity of a regular Transformer and PoET when generating a protein sequence conditioned on a fixed number of tokens from other sequences in the same protein family. Protein families consist of UniRef50 sequences, and both models perform much better than a profile HMM baseline (Appendix Figure 12).

but by a smaller margin. Next, we finetuned both models with 16K and 32K context lengths for 50K and 30K steps respectively. While the perplexity of the Transformer improved significantly at the 16K and 32K context lengths, the Transformer still does not generalize to context lengths beyond the training context length, and underperforms all variants of PoET at all context lengths. The Transformer also underperforms PoET for variant fitness prediction (Appendix H).

## 6 PoET generates novel sequences that preserve structure within a protein family

PoET's state-of-the-art performance on variant fitness prediction demonstrates that it assigns higher likelihood to regions of sequence space with high fitness variants. This suggests that PoET can be used to directly generate high fitness variants belonging to a protein family by conditioning PoET on sequences from the protein family and sampling from the resulting conditional distribution. Direct generation of variants makes the exploration of higher order mutants computationally tractable as sequence space grows exponentially large with number of mutations, making it impossible to explore this space by scoring all such variants.

To evaluate PoET's ability to generate sequences from specific protein families, we sampled 1,000 sequences each from PoET conditioned on homologs from two protein families, phage lysozymes (PF00959) and chorismate mutases with activity in E. coli (Appendix K.2), and examined the novelty of the generated sequences and their predicted structural conservation. Sequence novelty was measured as the max sequence identity to any natural sequence (of the same protein family). Predicted

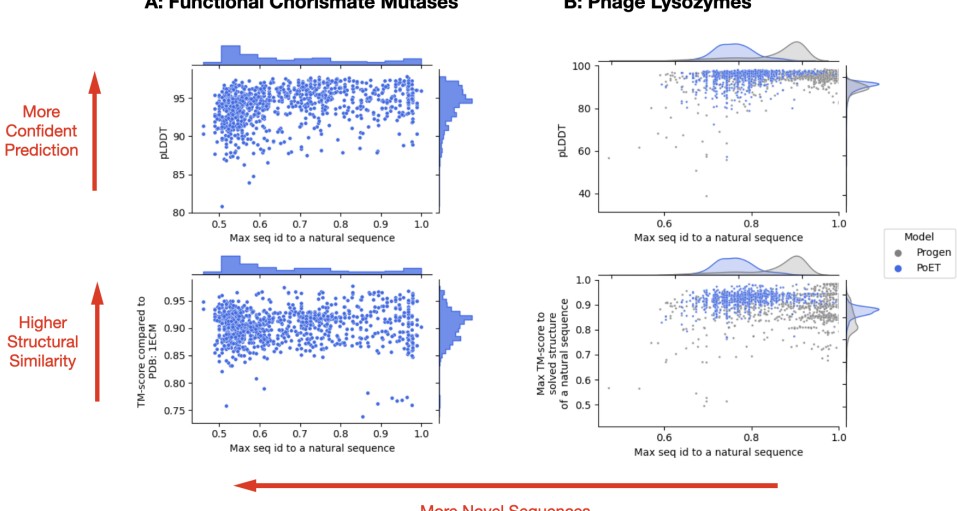

Figure 5: Sequence novelty and predicted structural conservation of (A) functional chorismate mutases generated by PoET and (B) phage lysozymes generated by PoET and ProGen. PoET generates diverse sequences (50-100% seq id to a natural sequence) while preserving 3D structure within the protein family (TM-score $> 0.8$ to a structure of a natural sequence and pLDDT $> 90$).

structural conservation was measured as the max TM-score between the predicted AlphaFold2 [39] structure of the generated sequence and the solved structure of any natural sequence. For both protein families, PoET generates high diversity sequences that are predicted with high confidence to be structurally similar, supporting the quality of sequences generated by PoET (Figure 5).

Furthermore, when tasked to generate phage lysozymes, we find that sequences generated by PoET are more diverse and better preserve structure at similar levels of sequence novelty than sequences generated by ProGen [8] (Figure 5B), a 1.2B parameter model fine-tuned specifically for generating phage lysozymes. As many sequences generated by ProGen have been experimentally confirmed to be functional phage lysozymes, these results suggest that PoET generates functional sequences with even greater diversity even without fine-tuning. Since fine-tuning ProGen is critical for achieving good variant effect prediction [8], and a version of ProGen fine-tuned on chorismate mutases is not available, we did not compare PoET and ProGen for generating novel chorismate mutases.

# 7 Conclusion

PoET is a Transformer-based autoregressive generative model of whole protein families. By framing family generation as a sequence-of-sequences generation problem, we are able to train across tens of millions of protein sequence clusters to encode fundamental rules of protein evolution into the PoET model. This enables the model to generalize to protein families unseen during training and extrapolate from small numbers of conditioned-upon sequences. The sequence-of-sequences generative framework allows PoET to be used as a retrieval-augmented language model, generating new sequences conditioned on a set of sequences representing the family or other properties of interest. We demonstrate that PoET improves over other protein language models and evolutionary sequence models for variant fitness prediction across a wide range of deep mutational scanning datasets. PoET also enables efficient sequence generation and the generative distribution can be controlled via conditioning. Phage lysozyme- and chorismate mutase-like sequences sampled from PoET are novel and predicted to fold with high confidence. PoET can be backed by other sequence databases and naturally improves as databases grow without the need for retraining. We anticipate that PoET will become a fundamental component of ML-enabled protein design in the future.

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

# A  Detailed description of `TieredTransformerDecoderLayer`

---

**Algorithm 1** `TieredTransformerDecoderLayer`

---

**Require:** representations for a sequence of sequences $h_{i,j} \in \mathbb{R}^d, i \in 1..N, j \in 1..L_i$

  First, apply causal self-attention to each sequence individually

  $f \leftarrow \texttt{LayerNorm}(h)$

  $q_{i,j} \leftarrow \texttt{RoPE}(\texttt{Linear}(f_{i,j}), j) \; \forall i, j$

  $k_{i,j} \leftarrow \texttt{RoPE}(\texttt{Linear}(f_{i,j}), j) \; \forall i, j$

  $v \leftarrow \texttt{Linear}(f)$

  $f_i \leftarrow f_i + \texttt{CausalAttention}(q_i, k_i, v_i) \; \forall i$

  Next, apply causal self-attention to all sequences together

  $g_{i,j} \leftarrow f_{i,j} \; \forall i, j$

  $g \leftarrow \texttt{LayerNorm}(g)$

  $q_{i,j} \leftarrow \texttt{RoPE}(\texttt{Linear}(g_{i,j}), j) \; \forall i, j$

  $k_{i,j} \leftarrow \texttt{RoPE}(\texttt{Linear}(g_{i,j}), j) \; \forall i, j$

  $v \leftarrow \texttt{Linear}(g)$

  $g \leftarrow g + \texttt{CausalAttention}(q, k, v)$

  Finally, the feedforward layer

  $g' \leftarrow \texttt{LayerNorm}(g)$

  $h' \leftarrow \texttt{GELU}(\texttt{Linear}(g')), h' \in \mathbb{R}^{4d}$

  $g' \leftarrow g + \texttt{Linear}(h')$

**output** $g' \in \mathbb{R}^d$

---

# B  Additional Training Details

## B.1  Data

To find homologs in UniRef50 using Diamond, we used the following command, which defines an all-against-all search:

```
diamond blastp -q uniref50.fasta -d diamond/uniref50 -f 6 -header -k 200000
-max-hsps 1 -e 0.001 -p 96 -o output.tab
```

The command returns, for each sequence in UniRef50, a set containing all its putative homologs in UniRef50. We call each such set a "Diamond-UniRef50 Cluster". Diamond was used over other homology search tools due to its high performance (>100x speed of BLAST).

To form a training example, sequences are sampled without replacement from a "Diamond-UniRef50 Cluster" until the total number of tokens reaches a predetermined limit, and then concatenated to form a sequence-of-sequences. Each sampled UniRef50 sequence is replaced with a UniRef100 sequence by sampling a random UniRef100 sequence from the same UniRef50 cluster as the UniRef50 sequence being replaced. The UniRef100 sequences are randomly sampled with weight inversely proportional to the size of the UniRef90 clusters they belong to. Each UniRef100 sequence is sampled at most once in each sequence-of-sequences.

As a final data augmentation, the order of the tokens in a sequence-of-sequences is reversed with probability 50% (i.e. *all* sequences in a sequence-of-sequences are ordered from either N-terminus to C-terminus only or C-terminus to N-terminus only). This augmentation has been shown to improve the performance of other protein language models [11] .

Following this sampling procedure, the order of sequences in a sequence-of-sequneces is random, which promotes order invariance. To evaluate the sensitivity of PoET to the order of sequences, we compare heldout sequence perplexities after conditioning on 10 sequences ordered either 1) randomly as described above, 2) shortest-to-longest, or 3) longest-to-shortest. We also compare the full joint log likelihoods of these sequences of 10 sequences with the same orderings (Table 2).

We find that sequence ordering has little-to-no effect on next sequence perplexity and joint log likelihoods. Ordering sequences shortest-to-longest changes next sequence perplexity by only 0.00753 on average and has a relative difference of $< 0.5\%$ from random for both next sequence perplexity and joint log likelihood.

Table 2: Comparison of (a) next sequence perplexities and (b) joint log likelihoods $\log p(s_1, s_2, \ldots, s_n)$, of sequences of 10 sequences with different orderings from heldout Uniref50 sequence clusters. We report mean perplexity and joint log likelihood as well as mean difference from random and relative difference from random for shortest-to-longest and longest-to-shortest orderings with the standard deviation across clusters in parenthesis. Relative difference is calculated as $(a - r)/r$ where $a$ is the ordered value and $r$ is the value with random ordering.

(a) Perplexity of Next Sequence N=9184

| Sequence Order | Mean Perplexity | Mean Difference from Random | Mean Relative Difference from Random |
|---|---|---|---|
| Random | 7.44075 | - | - |
| Shortest-to-Longest | 7.44828 | +0.00753 (0.07686) | +0.00109 (0.01033) |
| Longest-to-Shortest | 7.44410 | +0.00336 (0.07608) | +0.00055 (0.01030) |

(b) Joint Log Likelihood N=9184

| Sequence Order | Mean Joint Log Likelihood | Mean Difference from Random | Mean Relative Difference from Random |
|---|---|---|---|
| Random | -8099.74973 | - | - |
| Shortest-to-Longest | -8123.18167 | -23.43149 (55.76313) | -0.00335 (0.00838) |
| Longest-to-Shortest | -8091.89462 | +7.85511 (52.92428) | +0.00105 (0.00746) |

## B.2 Optimizer and learning rate

We used the AdaFactor optimizer [40] with initial learning rate 1e-2, square root learning rate decay, and otherwise default parameters.

## B.3 Hyperparameters

Hyperparameters for PoET variations used in ablation experiments (§5.2.1) are summarized in Table 3.

Table 3: Hyperparameters for PoET models of different sizes.

| # Parameters | $N_{\text{Layers}}$ | Hidden Dim | $N_{\text{Head}}$ | Batch Size (# Tokens) |
|---|---|---|---|---|
| 57M | 6 | 768 | 12 | 500k |
| 201M | 12 | 1024 | 16 | 250k |
| 604M | 16 | 1536 | 24 | 250k |

## C  Homologous Sequence Retrieval Methods

We experimented with two methods for retrieving homologous sequences from UniRef100, JackHM-Mer [41] and MMseqs2 [42]. Both programs return an MSA of homologs. For JackHMMer, we used the MSAs provided by ProteinGym [11], which were constructed from UniRef100 Version 2104. For MMseqs2, we used the protocol defined by ColabFold [35], and retrieved homologs from UniRef100 Version 2202. The main advantage of the ColabFold protocol is that it is substantially faster than the ProteinGym protocol (minutes vs up to hours), whereas the ProteinGym protocol may be more sensitive.

We evaluated all relevant baselines and PoET with homologs retrieved by both protocols (Table 4). Note that in the main result table (Table 1), we report the results of each method with the best protocol based on validation set performance. We replicate the findings of Laine et al. [12], who show that GEMME performs similarly with both protocols, and also find that most models perform similarly regardless of the protocol. The notable exception is EVE, which performs significantly worse. This is

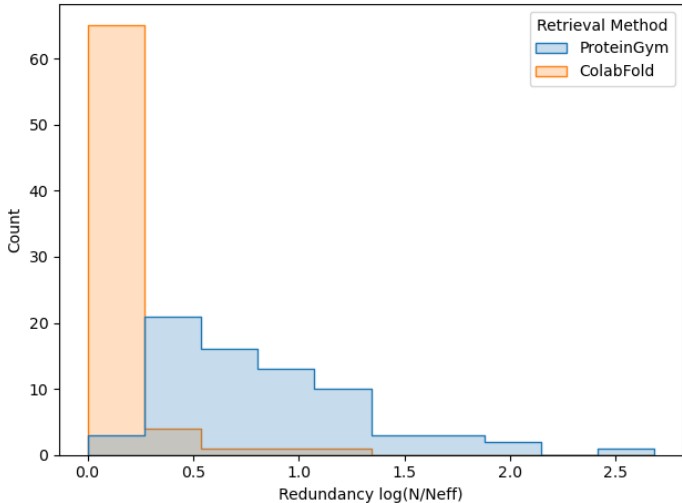

Figure 6: Redundancy, as measured by $N/N_{\text{eff}}$, of homologous sequence sets retrieved by the ProteinGym and ColabFold retrieval methods for target proteins in ProteinGym.

most likely because the MSAs returned by the ColabFold protocol are substantially less redundant (Figure 6). Thus they do not contain enough sequences to properly train EVE models, which are trained from scratch for each protein family and can contain millions of parameters. Removing redundancy filters in the protocol could improve results for EVE; we leave this to further research.

Given the poor performance of EVE with the ColabFold protocol, we did not also evaluate the EVE ensemble or TranceptEVE M/L with ColabFold, presuming that the performance of both would also be worse with ColabFold because they both depend on EVE. Furthermore, we did not evaluate the performance of Tranception with ColabFold because Tranception relies on homologs via the site independent model, which performs similarly with both protocols.

Table 4: Average Spearman correlation on ProteinGym subsets of baseline models and PoET using different methods for retrieving homologs.

| Model | Retrieval Method | Val | Subst. |
|---|---|---|---|
| Site independent | ProteinGym | **0.386** | **0.408** |
| | ColabFold | 0.371 | 0.404 |
| GEMME | ProteinGym | 0.416 | 0.457 |
| | ColabFold | **0.418** | **0.463** |
| EVE | ProteinGym | **0.423** | **0.443** |
| | ColabFold | 0.268 | 0.342 |
| MSA Transformer (ens.) | ProteinGym | 0.427 | **0.431** |
| | ColabFold | **0.440** | 0.423 |
| PoET (ens.) | ProteinGym | 0.463 | 0.474 |
| | ColabFold | **0.481** | **0.484** |

# D  Subsampling and Filtering of Retrieved Homologous Sequences

Even with their high sensitivity and specificity, homologous sequence retrieval methods are imperfect, and inevitably run into at least the following issues:

1. some sequences will be retrieved that do not share function with the target sequence, either due to false positives (i.e. sequences found that do not in fact share common ancestry), or because sequences have diverged in function despite their common ancestry [43]

2. the set of sequences retrieved will be biased towards those that have been chosen to be sequenced by humans, or by natural processes that may not reflect relative fitness [17]

To resolve the second issue, we adopt the sequence weighting scheme used by Hopf et al. [17], which has been shown to work well in many previous studies [4, 11, 13, 40]. The weighting scheme is detailed in Appendix D.5.

To address the first issue, existing methods have used sequence identity as a simple heuristic for filtering out irrelevant or misleading sequences. Laine et al. [12] apply a max dissimilarity filter of 0.8 sequence identity to remove sequences that are less likely to be homologous, and a max similarity filter of 0.98 sequence identity to remove sequences that are redundant with the target. Meier et al. [4] explore using a variety of max similarity filter thresholds and find that a threshold of 0.9 or 0.75 tends to be optimal, depending on the target. Both Laine et al. [12] and Meier et al. [4] also limit the total number of sequences in a subsample, which helps to limit the computational cost of inference, and ensemble the results from applying their model to multiple subsamples to obtain better predictions.

To determine the best way to subsample and filter homologous sequences for PoET, we evaluate the performance of the model on the ProteinGym validation set with (1) various max dissimilarity and (2) similarity thresholds for filtering, and (3) various context lengths (i.e. total number of tokens used to compute the conditional log-likelihood (Equation 3), including the tokens of the variant sequence being scored) for limiting the computational cost of inference.

### D.1   Maximum Sequence Dissimilarity

First, we explored the impact of the maximum sequence dissmilarity threshold. We evaluate PoET with (1) homologous sequences retrieved using either the ProteinGym or the ColabFold protocol, (2) a context length of 8192, and (3) various max sequence dissimilarity thresholds (Figure 7). When retrieving homologous sequences using the ProteinGym protocol, a threshold of 0.7 is optimal for both the validation and substitution benchmarks, and is significantly better than no filtering i.e. using a threshold of 1.0 (substitutions benchmark: $\Delta\bar{\rho} = 0.024, p < 0.01$). Interestingly, filtering was particularly helpful for prokaryotic datasets, suggesting that the ProteinGym protocol may be prone to finding false positives in this taxon. In contrast, for the ColabFold protocol, no filtering based on max dissimilarity is necessary; there is essentially no difference in performance when filtering with a threshold of 0.7.

### D.2   Maximum Sequence Similarity and Context Length

Next, using the optimal maximum sequence dissimilarity thresholds, we evaluated PoET while jointly varying the max similarity threshold $\in \{1.0, 0.95, 0.90, 0.70, 0.50\}$ and context length $\in \{6144, 12288, 24576\}$ (Figure 8). When averaged across datasets, we found that PoET generally performed better when some sequences are filtered i.e. max similarity threshold $< 1.0$, and with smaller context lengths (6144 or 12288). However, when looking at individual datasets (Figure 9), we find a wide variety of behaviors. On some datasets, increasing the context length consistently improves performance across max similarity thresholds and the optimal max similarity threshold is 1.0 (no filtering), whereas for others the opposite is true, and yet for others the behavior lies somewhere in between.

From these observations, we draw two main conclusions. First, although the average performance is lower at higher context lengths, the ability of PoET to improve in performance with increased context length on some datasets suggests that PoET is able to generalize to context lengths much longer than the training context length, and that decreased average performance at higher context lengths may be due to factors other than the model's ability to handle long context lengths. Weinstein et al. [44] suggest that optimal modeling of the sequences in an observed protein family may be *detrimental* to protein fitness prediction due to biases in the sequences we observe. Consequently, protein fitness prediction based on evolutionary models perform better when they do not or cannot model the family exactly e.g. via model misspecification. In the context of PoET then, limiting the max sequence similarity and context length when sampling sequences from the full set of retrieved homologs may

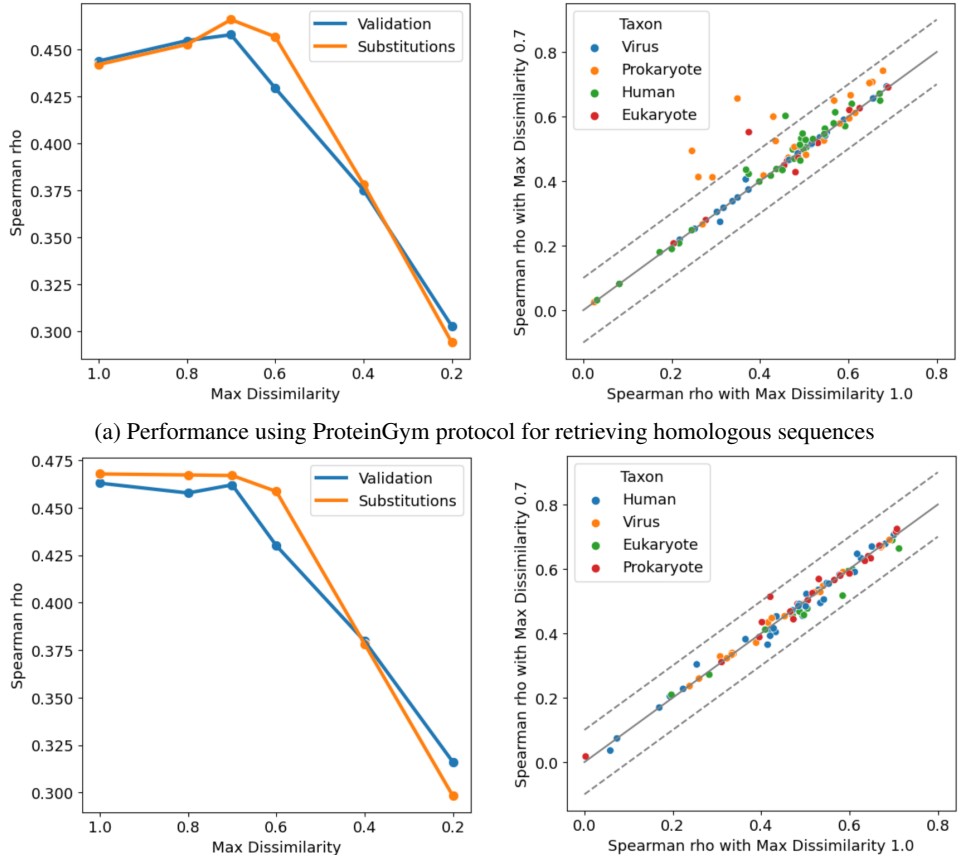

(a) Performance using ProteinGym protocol for retrieving homologous sequences

(b) Performance using ColabFold protocol for retrieving homologous sequences

Figure 7: **Optimal threshold for filtering homologous sequences by max dissimilarity to target sequence differs depending on retrieval method** *Left*: performance as a function of max dissimilarity threshold when using the ProteinGym protocol (top) and the ColabFold protocol (bottom). For ProteinGym, a threshold of 0.7 is optimal and significantly better than no filtering i.e. using a threshold of 1.0 (paired t-test on full substitutions benchmark: $p < 0.01$, effect size: $\Delta\bar{\rho} = 0.024$). For ColabFold, no filtering is necessary and there is no essentially no difference in performance when filtering with a threshold of 0.7. *Right*: comparison of Spearman correlations on datasets when filtering homologous sequences to 0.7 max dissimilarity vs not filtering, broken down by taxon. For ProteinGym (top), filtering improves performance particularly on Prokaryotic datasets, whereas for ColabFold (bottom), there is no clear pattern.

help avoid some of these biases while still providing the model information about the specific protein family of interest. Second, the large variance in the optimal max similarity threshold and context length on a per dataset level suggests that better sampling approaches, possibly dataset specific, may be an avenue for further improving protein fitness prediction.

### D.3 Ensemble over subsamples

Given the variance in best max sequence similarity and context length parameters on the validation datasets, we explored ensembling (Equation 4) over subsamples where the subsamples are drawn with different values for max similarity and context length. Table 5 compares (1) not ensembling, (2) ensembling with $N_{\text{ensemble}} = 15$ subsamples where the max sequence similarity and context length parameters are *fixed* to the best values on the validation set (Figure 8), and (3) ensembling subsamples with *varied* parameters (max similarity threshold $\in \{1.0, 0.95, 0.90, 0.70, 0.50\}$ and context length $\in \{6144, 12288, 24576\}$; 15 combinations in total). On the validation and substitutions benchmarks, we find that both types of ensembling improve performance. On the validation benchmark, the two ensembling methods perform similarly; fixed has a slight edge when using the ProteinGym protocol

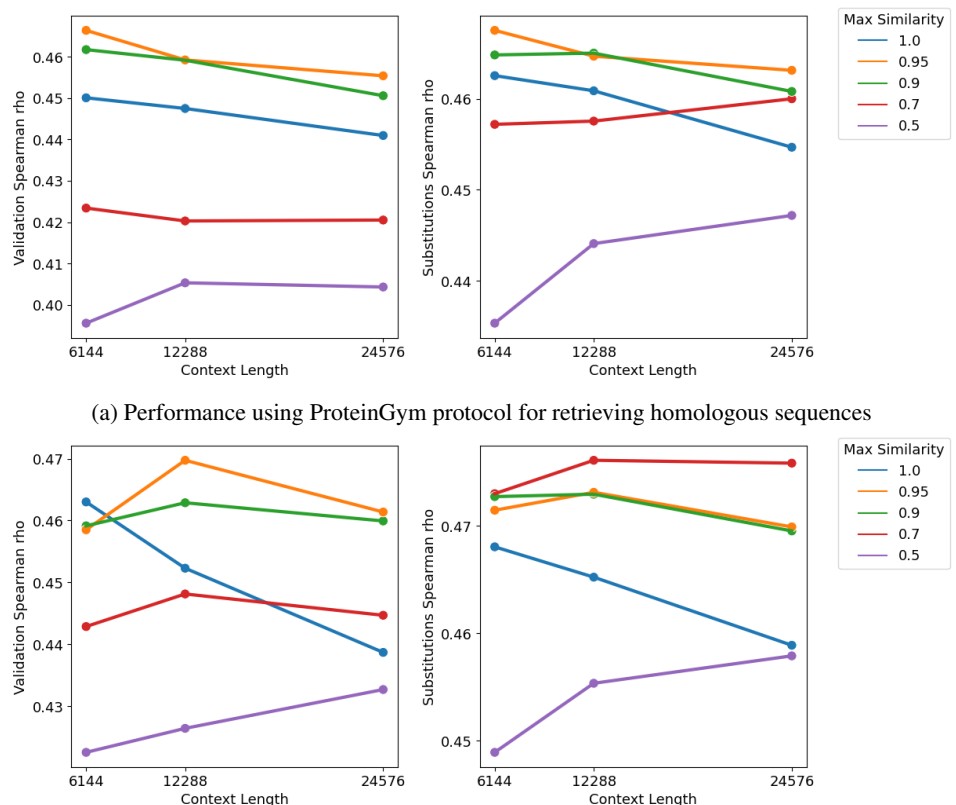

(a) Performance using ProteinGym protocol for retrieving homologous sequences

(b) Performance using ColabFold protocol for retrieving homologous sequences

Figure 8: **Averaged across datasets, PoET performs best with some filtering based on max similarity and at moderate context lengths.** Each line plot shows the performance of PoET on ProteinGym as the max similarity threshold and context length is varied. *Left*: Performance on the validation set. *Right*: Performance on the full substitutions benchmark.

Table 5: Average Spearman correlation between model scores and experimental measurements on ProteinGym for PoET when the homologous sequence subsampling parameters are fixed or varied across the ensemble, broken down by retrieval and sampling methods.

| Retrieval Method | Max Similarity and Context Length Parameters | Val | Substitutions | Indels |
|---|---|---|---|---|
| ProteinGym | Fixed (no ensemble) | 0.442 | 0.445 | 0.474 |
| | Fixed (15x ensemble) | 0.464 | 0.467 | **0.521** |
| | Varied (15 combinations) | 0.463 | 0.474 | 0.498 |
| ColabFold | Fixed (no ensemble) | 0.451 | 0.445 | 0.485 |
| | Fixed (15x ensemble) | 0.478 | 0.469 | 0.512 |
| | Varied (15 combinations) | **0.481** | **0.484** | 0.510 |

for retrieving homologs, and varied has a slight edge when using the ColabFold protocol. On the full substitutions benchmark, the varied method consistently performs better, and by a larger margin; this result may be less biased than the validation benchmark result because the parameters for the fixed method are tuned on the validation benchmark.

## D.4   Summary

Based on validation benchmark performance of the PoET variants explored in the above sections, for our final PoET model, we choose the following parameters

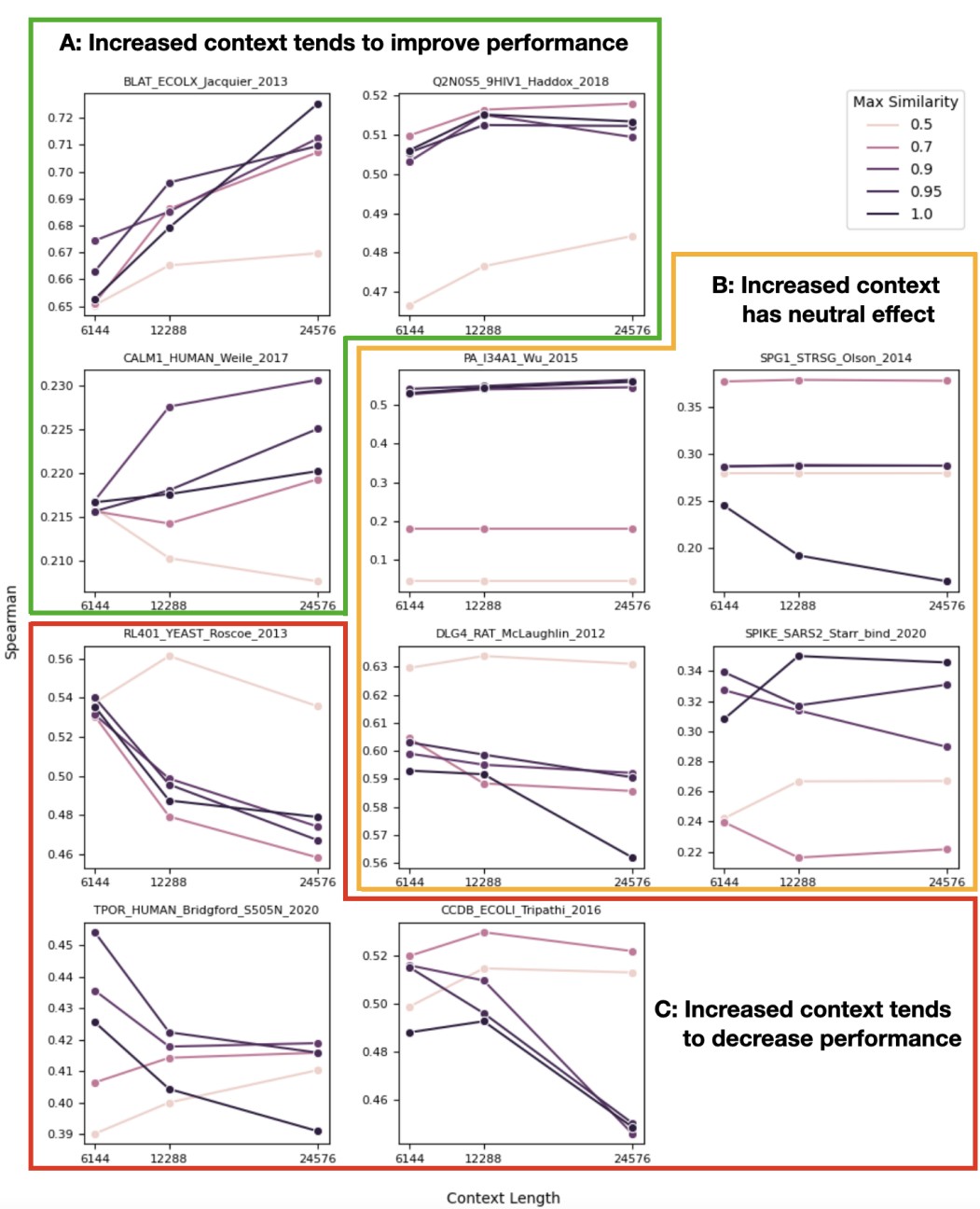

Figure 9: **Optimal max similarity threshold and context length varies across ProteinGym datasets.** Each line plot shows the performance on a validation dataset as the max similarity threshold and context length is varied. Datasets are grouped based on the effect of increasing context length for that dataset. *A*: Datasets for which increasing the context length tends to *improve* performance. *B*: Datasets for which increasing the context length tends to have a *neutral* effect on performance. *C*: Datasets for which increasing the context length tends to have a *negative* effect on performance.

1. Retrieve homologous sequences using the ColabFold protocol

2. Do not filter homologous sequences by max dissimilarity when subsampling the set of homologous sequences

3. Ensemble the predictions of PoET across subsamples of the full set of homologous sequences with max similarity thresholds $\in \{1.0, 0.95, 0.90, 0.70, 0.50\}$ and context lengths $\in \{6144, 12288, 24576\}$ (15 combinations total)

### D.5 Sequence Weighting During Homologous Sequence Sampling

We sampled homologous sequences using the sequence weighting scheme described in Hopf et al. [17]. Specifically, sequences are sampled from a probability distribution where a sequence's probability is inversely proportional to its number of neighbors in the MSA of the homologous sequences. A target sequence is considered a neighbor of a query sequence if the sequence identity to the query is greater than 0.8. The sequence identity is computed based on the alignment in the MSA, and gaps in the query sequence are ignored.

### D.6 Full length sequences vs aligned fragments

In the subsampling and filtering of homologous sequences experiments above, we condition on only the amino acids in the homologous sequences that are aligned to a position of the target sequence in the MSA of the homologous sequences i.e. a sequence in a subset $S_j$ (Equation 4) may *not* be the full length sequence as recorded in UniRef100, rather it is potentially a non-contiguous subsequence of the UniRef100 sequence. We find that using only these "aligned fragments" performs similarly to using the full length sequences (Table 6), and use the former over the latter as the former can be derived directly from the MSA.

Table 6: Average Spearman's rank correlation on the validation and substitutions benchmarks using full length sequences vs aligned fragments sampled from MSAs of homologous sequences generated by the ColabFold protocol.

| Method | Val | Substitutions |
|---|---|---|
| Full Length Sequences | 0.464 | 0.474 |
| Aligned Fragments | **0.466** | **0.479** |

## E MSA Depth

MSA depth measures the amount of sequence information the MSA contains about the target protein. We use the same definition and low/medium/high categorization as Notin et al. [11]. At a high level, the MSA depth $N_{\text{eff}}/L$ is computed as the number of non-redundant sequences $N_{\text{eff}}$ (at 0.8 sequence identity for non-viral proteins, and at 0.99 sequence identity for viral proteins) divided by the sequence length $L$. The MSA depth categories thresholds are - Low: $N_{\text{eff}}/L < 1$; Medium: $1 < N_{\text{eff}}/L < 100$; High: $N_{\text{eff}}/L < 100$.

Although the MSA depth changes based on the method used to retrieve homologous sequences, we categorize proteins/datasets using the ProteinGym MSAs (1) in order to maintain consistency with the categorizations in the ProteinGym benchmark, which allows for easier comparison with existing work, and (2) because the MSA depth is highly correlated between the retrieval methods considered in this paper (Figure 10).

## F Protein Variant Fitness Prediction Baselines

### F.1 Overview of Baselines

We compare PoET to the following baseline models for protein variant fitness prediction, which includes the best existing alignment-based models, unconditional and conditional protein language models, and hybrid models that combine multiple approaches models:

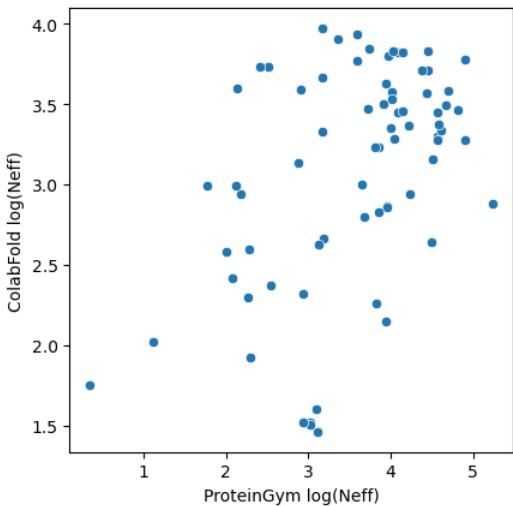

Figure 10: Comparison of MSA depth ($N_\text{eff}$) of MSAs of homologous sequences retrieved by the ProteinGym and ColabFold protocols for target proteins in ProteinGym ($\rho = 0.4, p < 0.001, n = 72$).

### F.1.1 Alignment-based Models

Alignment-based models first compute a MSA of homologous sequences of a given protein family, and compute sequence likelihoods based on MSA statistics or models trained on the MSA.

**Site Independent Model** The simplest alignment-based model; sequence likelihoods are computed from the frequency of amino acids at each position of the MSA, considered independently from other positions of the MSA.

**GEMME [12]** is a fast, scalable, and simple algorithm for predicting variant effect that models the interdependencies between different positions in the MSA by considering the evolutionary tree relating the sequences in the MSA. It is among the best performing baseline models while also requiring substantially less compute than comparably performing alternatives, but it cannot score indels.

**EVE [13] (ensemble)** is an ensemble of five Bayesian variational autoencoder trained on the MSA.

### F.1.2 Unconditional PLMs

Unconditional PLMs are large language models trained on protein databases spanning many protein families e.g. UniRef [23], using either the causal or masked language modeling objective.

**ESM-1v [4] (ensemble)** is an ensemble of five 650M parameter Transformer-based masked language models trained on UniRef90.

**ProGen2 [5] (ensemble)** is an ensemble of five Transformer-based causal language models ranging between 150M and 6.4B parameters and trained on UniRef90 and BFD [45].

**Tranception L (no retrieval) [11]** is a 700M parameter Transformer-based causal language model trained on UniRef100.

### F.1.3 Conditional PLMs

Conditional PLMs are large language models trained on sets of homologous protein sequences spanning many protein families.

**MSA Transformer [24] (ensemble)** is a 100M parameter masked language model trained on MSAs of all UniRef50 sequences. MSAs are generated using HHblits [46].

### F.1.4 Hybrid

Hybrid models combine predictions from different types of models.

**Tranception L [11]**  combines predictions from Tranception L (no retrieval) and a site independent model in a manner that allows novel indels to be scored by aligning the sequence to be scored to the MSA used by the site independent model.

**TranceptEVE [14]**  combines predictions from Tranception and EVE in a manner that allows novel indels to be scored by using EVE to "define a family-specific prior distribution over amino acids at each sequence position that is independent from the particular protein sequence". It is the best performing baseline method.

### F.2  Methodology

We run all baselines using the same methodology as Notin et al. [11, 14], with two exceptions:

1. **Homologous Sequence Retrieval Method** We explored using two methods, the ProteinGym and ColabFold protocols, for retrieving the homologous sequences used by baselines (Appendix C, Table 4), and compare all methods using the best homologous sequence retrieval method (Table 1).

2. **Site Independent Model** We present results for the site independent model using sequence weighting (Appendix D.5) and filtering sequences with more than 0.8 (in terms of sequence identity) dissimilarity to the target sequence. These are common preprocessing steps used to improve the performance of other models for protein variant fitness prediction (Appendix D), and are the same parameters used to define the retrieval-based site independent prior in Tranception [11] and TranceptEVE [14]. We believe that this variation of the site independent model better reflects the model class's ability to predict protein variant fitness.

## G  Ensembling PoET with Baseline Methods for Protein Variant Fitness Prediction

To ensemble PoET with a baseline method for protein variant fitness prediction, we compute a weighted sum of the score predicted by PoET and by the baseline method i.e.

$$\hat{F}_{\text{PoET+Baseline},i} = \hat{F}_{\text{PoET},i} + \alpha \times \hat{F}_{\text{Baseline},i} \tag{5}$$

where $\hat{F}_{\text{Baseline},i}$ is the score predicted by the baseline method, $\hat{F}_{\text{PoET},i}$ is the score predicted by PoET, and $\hat{F}_{\text{PoET+Baseline},i}$ is the ensemble prediction score.

The weight $\alpha$ is determined by optimizing performance over the ProteinGym validation set using ternary search over the range $[1\text{e-}3, 1\text{e}4]$.

Table 7: **Average Spearman correlation between model scores and experimental measurements on ProteinGym for various ensembles of PoET and baseline methods.** Performance of TranceptEVE L, the best baseline method, and PoET alone are also shown for reference.

| Model A | Model B | Val | Substitutions | Indels |
|---|---|---|---|---|
| TranceptEVE L | N/A | 0.470 | 0.471 | 0.466 |
| PoET (ens., ProteinGym Protocol) | N/A | 0.463 | 0.474 | 0.498 |
| PoET (ens., ColabFold Protocol) | N/A | **0.481** | **0.484** | **0.510** |
| | GEMME | 0.477 | **0.488** | N/A |
| | EVE (ens.) | 0.485 | 0.487 | N/A |
| PoET (ens., ProteinGym Protocol) | Tranception L (no retrieval) | 0.471 | 0.473 | **0.512** |
| | MSA Transformer (ens.) | 0.476 | 0.472 | N/A |
| | Tranception L | 0.480 | 0.475 | 0.510 |
| | TranceptEVE L | **0.488** | 0.486 | 0.504 |
| | GEMME | 0.487 | **0.496** | N/A |
| | EVE (ens.) | **0.495** | 0.493 | N/A |
| PoET (ens., ColabFold Protocol) | Tranception L (no retrieval) | 0.485 | 0.484 | **0.525** |
| | MSA Transformer (ens.) | 0.487 | 0.484 | N/A |
| | Tranception L | 0.489 | 0.483 | **0.525** |
| | TranceptEVE L | 0.494 | 0.492 | 0.521 |

# H  Effect of Architecture on Protein Variant Fitness Prediction

Figure 11 compares the performance of the regular RoPe-based Transformer and PoET on the ProteinGym validation benchmark. We use homologous sequences retrieved using the ColabFold protocol, and explore various max similarity thresholds and context lengths as in Appendix D.2. PoET consistently outperforms the Transformer. It also extrapolates to context lengths longer than the training context length (8K), whereas the Transformer does not.

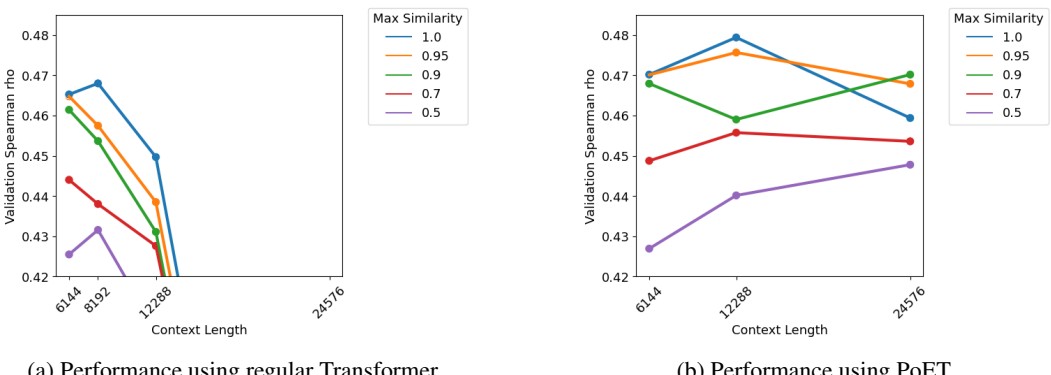

(a) Performance using regular Transformer.      (b) Performance using PoET.

Figure 11: Performance of regular RoPE-based Transformer and PoET on ProteinGym validation benchmark with homologous sequences retrieved using the ColabFold protocol and various max similarity thresholds and context lengths.

# I  Comparison of Models for Generating Sequences From a Protein Family

To benchmark PoET's ability to generate sequences from a protein family, we evaluated the mean perplexity of a protein sequence conditioned on a fixed number of tokens from other sequences in the same protein family across all "Diamond-UniRef50 clusters" (Appendix B) in the validation set. PoET significantly outperforms two baseline models, a profile HMM and a regular Transformer (Figure 12).

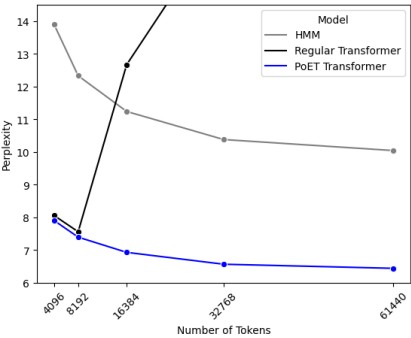

Figure 12: Comparison of the perplexity of a profile HMM, regular Transformer trained with 8K context length, and PoET trained with 8K context length when generating a protein sequence conditioned on a fixed number of tokens from other sequences in the same protein family. Protein families consist of UniRef50 sequences. The profile HMM is trained on MSAs inferred using `mafft` [47].

**PoET effectively models indel rich protein families**  The UniRef50 clusters in the validation set for perplexity evaluation are highly indel rich. An alignment for an exemplar cluster shows that there are large insertions and high diversity between sequences (Figure 13a). A histogram of the columns by percent gap shows that highly gappy columns are the most common (Figure 13b, left). Across all of the validation clusters (Figure 13b, right), this trend remains true, with alignments becoming

more indel rich when more sequences (longer context lengths) are considered. PoET achieves low perplexities on these heldout sequences, showing that it is a good generative model of indel rich families.

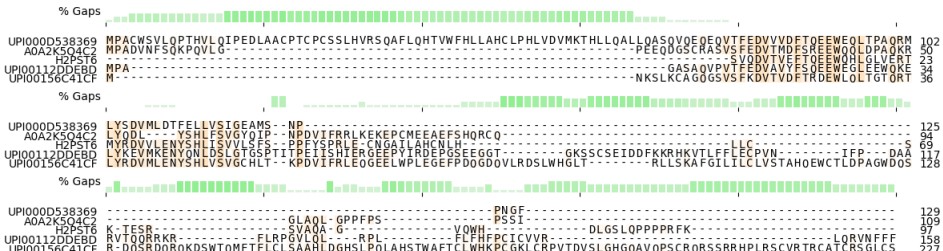

(a) Typical example of a MSA of a UniRef50 cluster. Bar plot above the MSA shows the percentage of gaps in each corresponding MSA column.

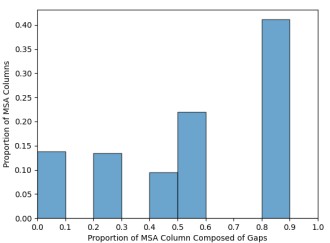 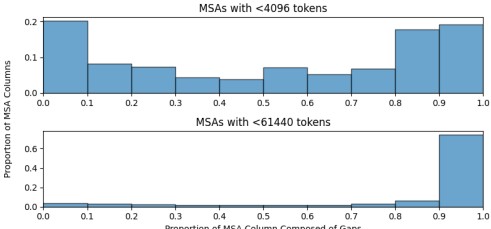

(b) (Left) Distribution of percentage of gaps in MSA columns for the example MSA above. (Right) Same distribution for all MSAs of UniRef50 clusters in the validation set with <4096 tokens (top) and <61440 tokens (bottom). Larger MSAs have more novel indels.

Figure 13: **MSAs of UniRef50 protein clusters reveal that these families contain many novel indels**

## J  Inference Speed of PoET vs TranceptEVE L

In the section, we compare the time it takes for PoET and TranceptEVE to predict the fitness of a set of protein variants on A100 GPUs. We begin by outlining the major steps required to perform inference for each method. For PoET, there are two main steps:

1. Retrieve homologs of the target protein using the ColabFold protocol
2. Compute the conditional log-likelihood of each variant given subsamples of the homologs. Do this for 15 subsamples and ensemble the results.

For TranceptEVE, there are five main steps:

1. Retrieve an MSA of homologs of the target protein using the ProteinGym protocol
2. Use the MSA to compute the site independent log prior
3. Train 5 EVE models on the MSA
4. Use the 5 EVE models to compute the EVE log prior
5. Compute the likelihood of each variant by combining scores from the Tranception protein language model, the site independent log prior, and the EVE log prior

Since TranceptEVE requires training new models and PoET does not, PoET can score many variants before TranceptEVE even finishes training. For simplicity of analysis, we establish a lower bound on the number of variants that can be scored by PoET before a single variant can be scored by TranceptEVE by computing the number of variants that can be scored by PoET in the time it takes to train one of the EVE models required for TranceptEVE. The homolog retrieval time is irrelevant for this analysis since both methods require a homolog retrieval step, although we note that the ColabFold

protocol used to retrieve homologs by PoET is substantially faster than the ProteinGym protocol used by TranceptEVE.

We find that independent of sequence length, on a single A100 GPU, PoET is able to score approximately 50,000 variants in the time it takes to train one EVE model. Furthermore, since variant scoring with PoET can be parallelized whereas training an EVE model cannot, inference with PoET can be sped up further by a factor of the number of A100 GPUs available. This allows up to hundreds of thousands of variants to be scored by PoET with a reasonable amount of compute before TranceptEVE is able to score a single variant.

# K  Function-specific variant fitness prediction and sequence generation via prompt engineering

## K.1  Function-specific variant fitness prediction

Thus far, we have evaluated the ability of PoET to predict the "general" fitness of a protein variant, where "general" fitness can refer to *any* property that is related to the function of the protein. But in practice, we are generally interested in optimizing *specific* properties of a protein. Since many properties of interest are correlated and together contribute to the general fitness of a protein (e.g. a more thermostable variant is also more likely to have higher expression or enzyme activity), optimizing general fitness is likely to optimize the true properties of interest, albeit indirectly. PoET is a useful tool for protein engineering because it is able to successfully predict general fitness by conditioning on and inferring evolutionary constraints from a diverse set of homologs of the target protein. This raises a natural question - can PoET be used to optimize *specific* properties of interest by conditioning on only the subset of relevant homologs that are known to or are predicted to display the specific properties of interest? It is difficult to answer this question in its full generality, since the process of selecting the subset of relevant homologs must be tailored to the specific target protein of interest, and there is an inevitable trade-off between learning from a smaller set of homologs that are more relevant, and a larger set of homologs from which there is more data to learn.

As a proof of concept, we demonstrate that PoET is able to learn function-specific evolutionary constraints for the target protein and property of interest in the chorismate mutase indels dataset [9] from ProteinGym. The dataset contains measurements of the catalytic activity, in E. coli, of 1130 natural chorismate mutase sequences, and 1618 designed chorismate mutase variants. The natural sequences are comprised of the target protein, a chorismate mutase found in E. coli, and homologs of the target protein found using the PSI-BLAST program for sequence search. The designed variants were selected by Monte Carlo sampling from a Potts model trained on an MSA of the natural sequences. This data presents an ideal scenario for selecting the subset of most relevant homologs; we can simply select the subset of natural homologs that are measured to be functional. In the absence of such data, we could instead use predictions from another model, or other relevant known attributes of the sequences e.g. if we would like to optimize for activity at high temperatures, we may select only the homologs from thermophiles.

On the chorismate mutase dataset, we find that the catalytic activity of designed chorismate mutase variants is better predicted when PoET is conditioned on only the subset of functional natural sequences rather than all natural sequences ($\Delta\rho = 0.2$, Figures 14b, 14d). In fact, PoET conditoned on functional natural sequences even outperforms fully supervised methods, including a Gaussian process trained on mean embeddings from a BERT-like protein masked language model ($\Delta\rho = 0.06$, Table 8). Such embeddings have been shown to be highly predictive of a variety of protein properties [20, 48, 49] and provide a strong baseline. These fully supervised methods are trained on more data than PoET because they train on the measured catalytic activities of all the natural sequences, whereas PoET is simply conditioned on positively labeled natural sequences and does not have access to the measured activities. This enables PoET to be used with assays that only measure binary endpoints rather than continuous values.

Table 8: Spearman correlation between measured and predicted fitness of designed chorismate mutase variants.

| Method | Spearman Correlation |
|---|---|
| Ridge regression trained on one hot encodings | 0.45173 |
| Gaussian process trained on mean embeddings from a BERT-like masked protein language model | 0.53658 |
| PoET conditoned on functional natural sequences | 0.56772 |

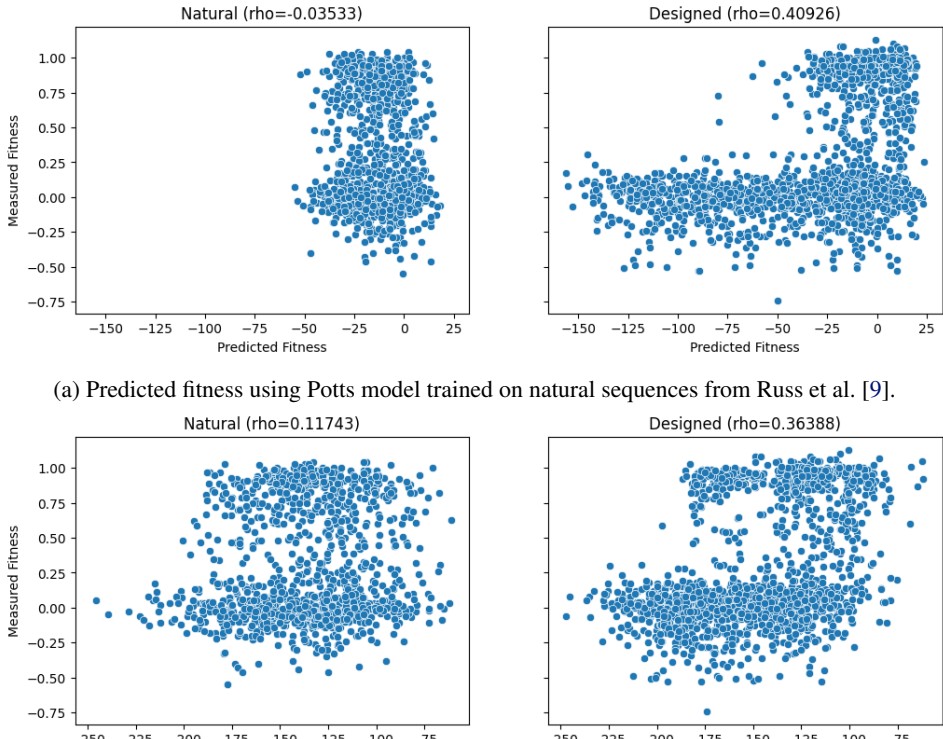

(a) Predicted fitness using Potts model trained on natural sequences from Russ et al. [9].

(b) Predicted fitness using PoET conditioned on natural sequences. Natural sequences were retrieved using PSI-BLAST to search for homologs of the target protein [9].

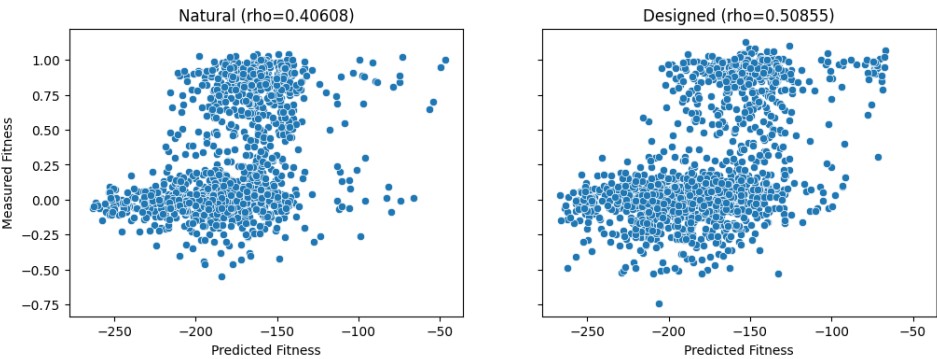

(c) Predicted fitness using PoET conditoned on homologs of the target protein retrieved by the ColabFold protocol.

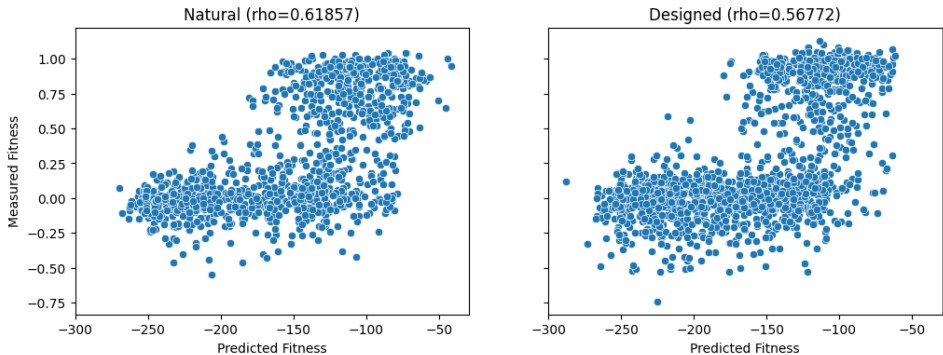

(d) Predicted fitness using PoET conditoned on functional natural sequences. Natural sequences with measured fitness >0.42 are considered functional by Russ et al. [9].

Figure 14: Measured fitness vs predicted fitness of (*left*) natural and (*right*) designed chorismate mutase enzymes from Russ et al. [9].

### K.2 Function-specific sequence generation

Conditioned on the functional natural sequences, we used PoET to generate 1000 novel putative chorismate mutases using nucleus sampling [50] with $p = 0.9$.

Figure 17 visualizes the sequence space spanned by natural chorismate mutases, chorismate mutases functional in E. coli, and sequences sampled from PoET when conditioned on the latter sequences. We find that the sequences sampled from PoET are concentrated in clusters around the known functional sequences, suggesting that the generated sequences sample from the functional subspace. The generated sequences are highly diverse and not simply a recapitulation of the natural sequences; the maximum sequence identity to any natural sequence is between 0.4 and 1.0, with a mode around 0.55 (Figure 15a). Nevertheless, the entropy at each position of an MSA of the functional natural sequences and an MSA of the generated sequence are closely matched (Figure 15b), indicating that the generated sequences reflect at least the first order evolutionary constraints of functional chorismate mutases.

High confidence (pLDDT>90) AlphaFold2 [35] predicted structures of generated sequences aligned to the experimentally solved structure of the target chorismate mutase (PDB: 1ECM) show that structure is preserved despite significant divergences in sequence identity; two examples are shown in Figure 16 and the full distributions of pLDDTs and TM-scores are shown in Figure 5A.

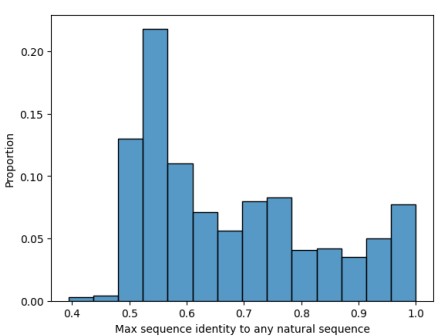

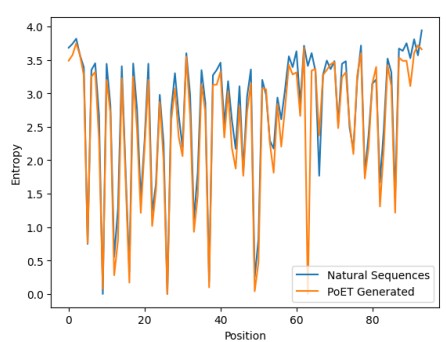

(a) Distribution of max seq id between generated and natural sequences

(b) Per position entropy of natural and generated sequences

Figure 15: Statistics of natural and generated sequences.

## L  Sampling methods for sequence generation experiments

When sampling sequences from PoET, we used nucleus sampling [50] with $p = 0.90$. When sampling sequences from ProGen, we tried nucleus sampling with $p \in \{0.25, 0.50, 0.75\}$, which are the three sampling strategies used by the authors of ProGen for lysozyme sequence generation. We found that when $p < 0.75$, the resulting libraries had low diversity ($> 50\%$ of generated sequences were duplicates). Therefore, we only present results with $p = 0.75$; under this setting, almost all generated sequences were unique.

## M  Limitations

**Orphan Proteins**  While ProteinGym includes DMS data on some proteins with low MSA depth, it does not include any orphan proteins that have *no* homologs. Simulating orphan proteins by e.g. removing all homologs from the training set is computationally intractable as it would require retraining large models and may be sensitive to homology search parameters. As a result, our experiments are inconclusive with respect to orphan proteins.

**Exhaustive scoring of all high order mutants**  Some methods such as GEMME can achieve similar albeit lower performance than PoET with substantially lower compute. However, no method is efficient enough to exhaustively score all variants with more than a couple of mutations. For

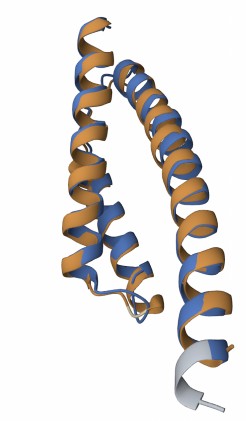 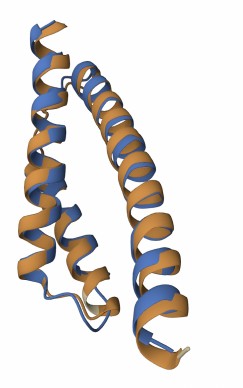

(a) Structure alignment with the highest scoring (according to PoET) generated sequence with maximum sequence identity of 0.8 to a natural sequence.

(b) Structure alignment with the highest scoring (according to PoET) generated sequence with maximum sequence identity of 0.5 to a natural sequence.

Figure 16: AlphaFold2 predicted structures of generated sequences aligned to the target chorismate mutase (PDB: 1ECM). Both alignments have RMSD $\leq$ 1.02 and TM Score $\geq$ 0.9.

exploring the space of higher order mutants, direct generation is far more effective, and is enabled by PoET via sampling.

**Limited compute for ablation studies**   Our ability to conduct ablations is ultimately constrained by compute resources. Ablation models were trained until validation set performance plateaued, but further training may have changed the results.

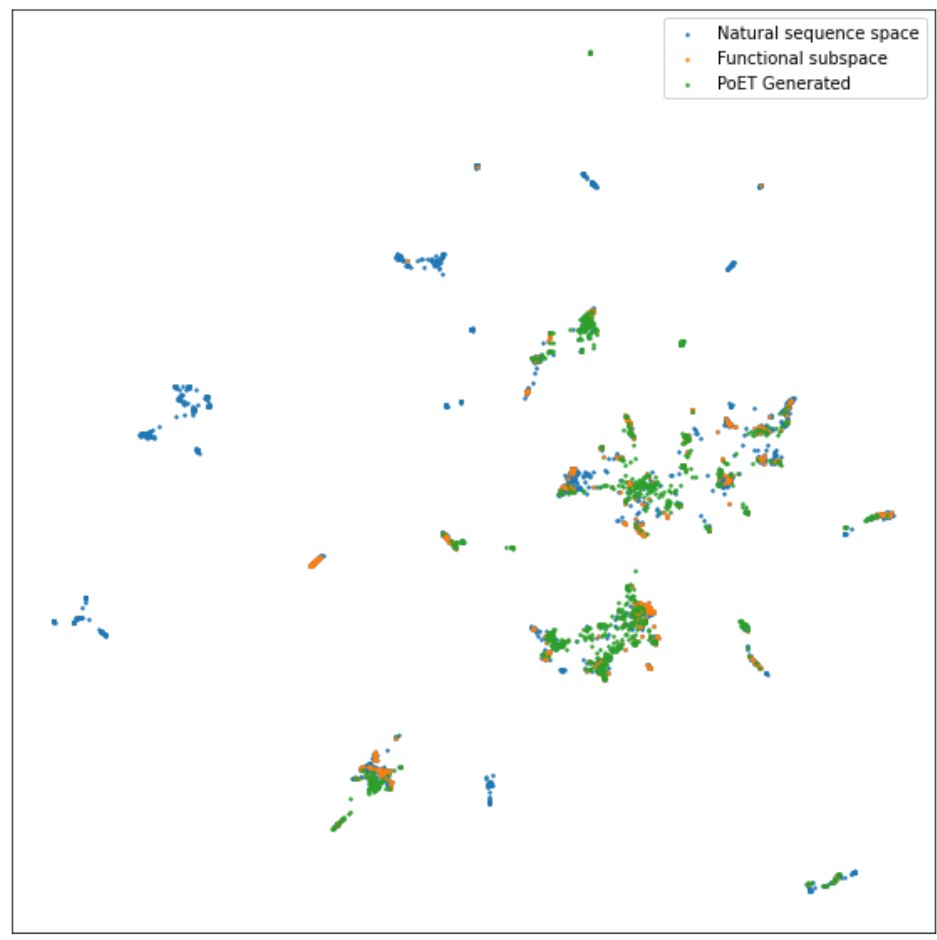

Figure 17: UMAP of natural chorismate mutases (blue), chorismate mutases functional in E. coli (orange), and sequences sampled from PoET when conditioned on the latter sequences (green). Sequences are compared using the Hamming distance metric.

# N  Supplementary Tables

Table S1: **Average Spearman correlation between model scores and experimental measurements on ProteinGym by MSA depth.** This is an extended version of the main result table (Table 1) that includes performance on the validation benchmark.

| Model type | Model name | Val | Low | Medium | High | All | Indels |
|---|---|---|---|---|---|---|---|
| | | | \multicolumn{4}{c}{Substitutions by MSA Depth} | | |
| Alignment-based model | Site independent | 0.386 | 0.417 | 0.404 | 0.411 | 0.408 | N/A |
| | GEMME | 0.418 | 0.445 | 0.449 | 0.522 | 0.463 | N/A |
| | EVE (ens.) | 0.435 | 0.414 | 0.441 | 0.498 | 0.448 | N/A |
| Uncond-itional PLM | ESM-1v (ens.) | 0.355 | 0.356 | 0.372 | 0.510 | 0.398 | N/A |
| | ProGen2 (ens.) | 0.397 | 0.357 | 0.416 | 0.448 | 0.411 | 0.407 |
| | Tranception L (no retrieval) | 0.396 | 0.377 | 0.399 | 0.429 | 0.401 | 0.430 |
| Cond-itional | MSA Transformer (ens.) | 0.440 | 0.372 | 0.421 | 0.477 | 0.423 | N/A |
| | PoET | **0.481** | **0.476** | **0.466** | **0.542** | **0.484** | **0.510** |
| Hybrid | Tranception L | 0.447 | 0.441 | 0.437 | 0.472 | 0.445 | 0.464 |
| | TranceptEVE M | - | - | - | - | - | 0.516 |
| | TranceptEVE L | 0.470 | 0.454 | 0.463 | 0.508 | 0.471 | 0.466 |
| | PoET +TranceptEVE L | **0.494** | **0.479** | **0.480** | **0.537** | 0.492 | **0.521** |

Table S2: **Average Spearman correlation between model scores and experimental measurements on ProteinGym by mutation depth.** Mutation depth is the number of substitutions a variant sequence has compared to the target sequence. Bolded values indicate the highest value in each column, with values above and below the double-line considered separately.

| Model type | Model name | Val | 1 | 2 | 3 | 4 | 5+ | All |
|---|---|---|---|---|---|---|---|---|
| | | | \multicolumn{6}{c}{Substitutions by Mutation Depth} | | | | |
| Alignment-based model | Site independent | 0.386 | 0.408 | 0.325 | 0.306 | 0.316 | 0.409 | 0.408 |
| | GEMME | 0.418 | 0.459 | 0.412 | 0.394 | 0.353 | 0.440 | 0.463 |
| | EVE (ens.) | 0.435 | 0.449 | 0.409 | 0.405 | 0.351 | 0.429 | 0.448 |
| Uncond-itional PLM | ESM-1v (ens.) | 0.355 | 0.396 | 0.309 | 0.203 | 0.165 | 0.253 | 0.398 |
| | ProGen2 (ens.) | 0.397 | 0.409 | 0.312 | 0.233 | 0.228 | 0.282 | 0.411 |
| | Tranception L (no retrieval) | 0.396 | 0.392 | 0.398 | 0.418 | 0.334 | 0.422 | 0.401 |
| Cond-itional | MSA Transformer (ens.) | 0.440 | 0.426 | 0.381 | 0.444 | 0.374 | 0.431 | 0.423 |
| | PoET | **0.481** | **0.483** | **0.456** | **0.473** | **0.404** | **0.449** | **0.484** |
| Hybrid | Tranception L | 0.447 | 0.441 | 0.427 | 0.440 | 0.370 | 0.461 | 0.445 |
| | TranceptEVE M | - | - | - | - | - | - | - |
| | TranceptEVE L | 0.470 | 0.471 | 0.445 | 0.456 | 0.389 | 0.464 | 0.471 |
| | PoET + TranceptEVE L | **0.494** | **0.492** | **0.465** | **0.478** | **0.411** | **0.471** | **0.492** |

Table S3: Average Spearman correlation between model scores and experimental measurements on ProteinGym by taxon.

| Model type | Model name | Val | Substitutions by Taxon | | | | |
| | | | Human | Other Eukary. | Prokary. | Virus | All |
|---|---|---|---|---|---|---|---|
| Alignment-based model | Site independent | 0.386 | 0.383 | 0.471 | 0.402 | 0.414 | 0.408 |
| | GEMME | 0.418 | 0.434 | 0.514 | 0.507 | 0.438 | 0.463 |
| | EVE (ens.) | 0.435 | 0.403 | 0.499 | 0.500 | 0.435 | 0.448 |
| Uncond-itional PLM | ESM-1v (ens.) | 0.355 | 0.417 | 0.441 | 0.502 | 0.256 | 0.398 |
| | ProGen2 (ens.) | 0.397 | 0.394 | 0.459 | 0.462 | 0.364 | 0.411 |
| | Tranception L (no retrieval) | 0.396 | 0.359 | 0.436 | 0.450 | 0.395 | 0.401 |
| Cond-itional | MSA Transformer (ens.) | 0.440 | 0.352 | 0.455 | 0.494 | 0.441 | 0.423 |
| | PoET | **0.481** | **0.445** | **0.524** | **0.526** | **0.479** | **0.484** |
| Hybrid | Tranception L | 0.447 | 0.413 | 0.503 | 0.478 | 0.429 | 0.445 |
| | TranceptEVE M | - | - | - | - | - | - |
| | TranceptEVE L | 0.470 | 0.436 | 0.518 | 0.514 | 0.454 | 0.471 |
| | PoET + TranceptEVE L | **0.494** | **0.450** | **0.538** | **0.536** | **0.486** | **0.492** |

# O    Supplementary Figures

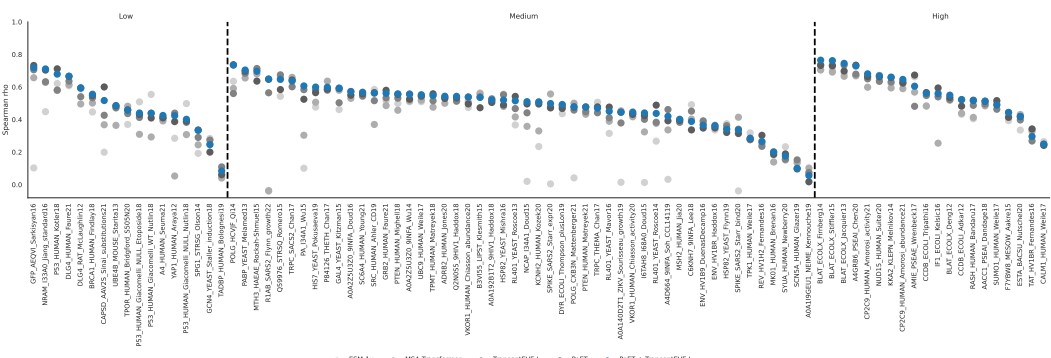

Figure S1: Model performance for each dataset in the ProteinGym substitutions benchmark.

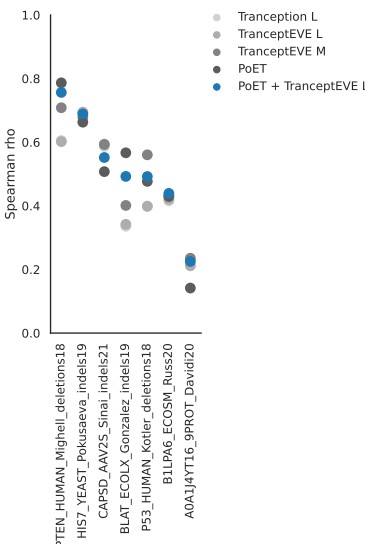

Figure S2: Model performance for each dataset in the ProteinGym indels benchmark.

