# OpenReview forum: "PoET: A generative model of protein families as sequences-of-sequences"
_NeurIPS.cc/2023/Conference — NeurIPS 2023 poster_

### Official Review · Reviewer_UQRE · 2023-07-04

**Soundness:** 3 good
**Presentation:** 2 fair
**Contribution:** 3 good
**Rating:** 5
**Confidence:** 3

**Summary:**

This paper proposed an autoregressive generation pre-trained model of protein families. The models are trained over the sequences-of-sequences being organized by a set of specific protein sequences. It utilized a shared in-sequence position encoder to capture conditioning among sequences in an order independent manner, and thus is able to generalize to large context lengths. This paradigm can help to explore the correlation among sub-sequences to improve the performance of generation, especially when lack of sufficient multiple sequence alignment supervised training data.

Experiments were conducted over DMS data sets to show the effectiveness of the proposed method.

**Strengths:**

1.Proposed an autoregressive generation model which is trained in the manner of sequences-of-sequences, which is easy to append more protein sequence and explore the correction among protein to guide the generation.

2.Shared absolute position encoder is employed in self-attention across different sub sequences to relieve the impact of sub-sequence ordering on next amino acid generation.

**Weaknesses:**

1.Although shared inter sequence position encoder can relive the impact of input sub-sequence order, the generation procedure is still order-dependent since Pr(next amino acid|s1, s2, ...). For example, if we arrange all short sequences at the begging of the combined sequence, it would be difficult to generated longer sequence such as insertion-based variant. It is better to explore such impact or validate whether the generation is robust to the input order by careful experimental study.

2.Lack of deep insight or analysis of the results, even when some results might be anti-intuitive, such as longer input sequence results in negative performance.

**Questions:**

1.Transformer’s attention mechanism still suffer from high time and memory consumption, please give the time and space complexity of the proposed transformer structure.
2.How to grantee the diversity of generated sequence? In autoregressive models, the maximum probability guided generation usually suffer from simplex result.
3.Generally, the hyper-parameters in ensembling will affect the evaluation performance. Does this mean a fair comparison in Table 1.
4.In ablation study, increasing the context length even results in negative performance. This is anti-intuitive. Is this phenomenon due to the decoding strategy (See question 2) or longer sequence weaken the effect of invert count sampling?
5.How to perform ensemble to fuse the PoET and the other baselines, what is the motivation of ensemble? It seems unfair to compare with other baselines w/o ensemble.

Typos:
213: All combinations of these are parameters are used

**Limitations:**

Please refer to weakness.

The organization and representation (typos) can be further improved. For example, it would be better to give a brief organization/introduction of experimental study at the beginning of Section 5.

---

> ### Author Rebuttal · Authors · 2023-08-10
>
> We thank the reviewer for their consideration of our paper, and for their questions and comments. We note that there seems to be some misunderstanding regarding certain aspects of our method and experiments, which we hope to address below in addition to the reviewer’s questions. We will also clarify these points in the revised manuscript.
>
> - Although shared inter sequence position encoder can relive the impact of input sub-sequence order, the generation procedure is still order-dependent since Pr(next amino acid|s1, s2, ...). For example, if we arrange all short sequences at the begging of the combined sequence, it would be difficult to generated longer sequence such as insertion-based variant. It is better to explore such impact or validate whether the generation is robust to the input order by careful experimental study.
>
> The model is not sensitive to the lengths and ordering of sequences in the way suggested. The model is able to generate novel insertions and the relative positional encodings only act as a prior on the alignment between sequences in multihead attention. The sequence clusters the model is trained and evaluated on have large numbers of indels between sequences and have high length variability. Please see our global response comment for more details.
>
> - Lack of deep insight or analysis of the results, even when some results might be anti-intuitive, such as longer input sequence results in negative performance.
>
> As pointed out by the other reviewers, we provide extensive analysis of the model and prompt construction methodology and discussion of these results in the appendix. That being said, we cannot explain every phenomena found in these experiments. In particular, we suspect that the model capacity and context length results are both related to the misspecification problem of variant function prediction and density estimation as discussed in [1] and Appendix D.2 Lines 399-412. Hopefully these will prove fruitful for us or others to explore in future work.
>
> - Transformer’s attention mechanism still suffer from high time and memory consumption, please give the time and space complexity of the proposed transformer structure.
>
> We utilize flash attention [2], so the model requires O(N^2) time and O(N) memory where N is the length of the sequence.
>
> - How to grantee the diversity of generated sequence? In autoregressive models, the maximum probability guided generation usually suffer from simplex result.
>
> Sampling from the model yields high diversity results and this diversity can be controlled using well known methods like nucleus sampling [3].
>
> - Generally, the hyper-parameters in ensembling will affect the evaluation performance. Does this mean a fair comparison in Table 1.
>
> There are no additional hyper-parameters in our ensemble. It is a simple average over each prompt. Other settings such as homolog retrieval method, sampling homologs, context length, etc, are tuned based on validation set performance (lines 204-208) and are not specific to the ensemble. Furthermore, several other methods use ensembles over large numbers of language models, whereas we only use one.
>
> - In ablation study, increasing the context length even results in negative performance. This is anti-intuitive. Is this phenomenon due to the decoding strategy (See question 2) or longer sequence weaken the effect of invert count sampling?
>
> We think this is probably due to the misspecification problem between variant fitness prediction and density estimation. Increasing the context length does generally improve the generative performance of the model (as measured by perplexity on heldout sequence clusters), but hurts performance on variant effect prediction.
>
> - How to perform ensemble to fuse the PoET and the other baselines, what is the motivation of ensemble? It seems unfair to compare with other baselines w/o ensemble.
>
> We simply average them together (Appendix H). This is why we report the PoET + TranceptEVE ensemble in a separate section, consistent with the TranceptEVE hybrid model, and separate from the other non-multi-method ensembles.
>
> - Typos: 213: All combinations of these are parameters are used
>
> We will correct this in the revised manuscript.
>
> [1] Weinstein, Eli, et al. "Non-identifiability and the Blessings of Misspecification in Models of Molecular Fitness." Advances in Neural Information Processing Systems 35 (2022): 5484-5497.
>
> [2] Dao, T., Fu, D. Y., Ermon, S., Rudra, A., & Ré, C. (2022). FlashAttention: Fast and Memory-Efficient Exact Attention with IO-Awareness. arXiv [Cs.LG]. Retrieved from http://arxiv.org/abs/2205.14135
>
> [3] Ari Holtzman, Jan Buys, Li Du, Maxwell Forbes, and Yejin Choi. The curious case of neural text degeneration. In International Conference on Learning Representations, 2020. URL https://openreview.net/forum?id=rygGQyrFvH.

---

> > ### Comment · Reviewer_UQRE · 2023-08-16
> >
> > Thanks for the authors giving detailed explanation and supplying more evidence to clarify my concerns, especially for the input order w.r.t. shared absolute position embedding. Except for the diversity of length distribution, this question talks about the sensitivity analysis of input order or the robustness of the proposed method. I think a better way to clarify this question would be to compare the generated sequences based on different input sequences, such as input sequences by concatenating shorter sequences at first vs. input sequences organized by random orders. This is due to the fact that the causal attention is not symmetric and would still be order dependent.
> >
> > If the generated sequences are either better than the baselines or independent of the input sequence order, I will increase my score.

---

> > > ### Author Response · Authors · 2023-08-18
> > > **Additional experiment added**
> > >
> > > We thank the reviewer for their additional clarification and helpful suggestions. We have performed the requested experiment and report next sequence perplexities and joint log likelihoods of sequences-of-sequences with random, shortest-to-longest, and longest-to-shortest orderings in our addendum comment (Addendum 2: on sequence ordering). We find that sequence ordering has little-to-no effect on next sequence perplexity or joint log likelihood with PoET, and that the results are significantly better than the baselines regardless of ordering. We also note that, in practice, we always use random orderings and orderings can even be marginalized during inference by sampling multiple random orderings. We will include these additional findings in the manuscript and hope this has sufficiently addressed the reviewer’s concern.

---

### Official Review · Reviewer_Uh6C · 2023-07-05

**Soundness:** 3 good
**Presentation:** 3 good
**Contribution:** 3 good
**Rating:** 5
**Confidence:** 3

**Summary:**

This paper proposes an autoregressive generative model (protein evolutionary transformer, PoET) of whole protein families. Current generative protein language models are either difficult to direct to produce a protein from a specific family of interest or must be trained on a large multiple sequence alignment (MSA) from the specific family of interest, making them unable to benefit from transfer learning across families. This model can incorporate new sequence information without retraining, generalize to large context lengths, and avoid issues related to conditioning on MSAs. They propose a novel Transformer layer that models order-dependence between tokens within sequences and order-independence between sequences. The advantages include that PoET can be used as a retrieval-augmented protein language model, 2) generate and score novel indels in addition to substitutions, and does not depend on MSAs of the input family, 3) extrapolate from short context lengths allowing it to generalize well even for small protein families.

**Strengths:**

- This paper proposes a novel Transformer layer that models order-dependence between tokens within sequences and order-independence between sequences.
- This paper provides a detailed analysis of experiments in the supplementary material.
- The proposed PoET outperforms existing protein language models and evolutionary sequence models for variant effect prediction in extensive experiments on the 94 deep mutational scanning datasets in ProteinGym.

**Weaknesses:**

- As the title stated, the core of this paper is a generative model of protein families, but they only do a downstream task, ie, fitness prediction. In this way, I suggest incorporating 'fitness prediction' into the title.
- The missing comparison with some fitness prediction baselines.


**Questions:**

1. Line 64, the contribution 'PoET can be sampled from and can be used to calculate the likelihood of any sequence efficiently.' sampled from which? Moreover, I don't think this is a contribution.
2. Line 197, why the log-likelihood of the variant can be used as the fitness predictor? What's the intuition behind this? or can you provide the related reference?
3. Line 227, Comparison to baselines. There are many methods for fitness prediction in directed evolution, such as CLADE and CLADE2.0. Why not compare with them?
4. In Table 1, What's the result of PoET if no ensemble?


**Limitations:**

This paper presents fitness prediction as the only downstream task of PoET. It is better to show more tasks to verify the effectiveness of PoET.

---

> ### Author Rebuttal · Authors · 2023-08-10
>
> We thank the reviewer for their positive assessment of our work and for their constructive comments and questions. These are addressed below.
>
> - As the title stated, the core of this paper is a generative model of protein families, but they only do a downstream task, ie, fitness prediction. In this way, I suggest incorporating 'fitness prediction' into the title.
>
> We consider generative evaluations of the model in terms of the perplexity of generating heldout sequences (Figure 4, 10) as well as examine samples from the model (Appendix M.2). Furthermore, we have expanded this analysis as discussed in the global response. That said, we do focus primarily on the evaluation of our model in terms of its ability to predict variant fitness. We will revise the manuscript to better emphasize this point.
>
> - Line 64, the contribution 'PoET can be sampled from and can be used to calculate the likelihood of any sequence efficiently.' sampled from which? Moreover, I don't think this is a contribution.
>
> What we mean by this is that samples can be efficiently drawn from the distribution over sequences modeled by PoET. PoET can also be used to calculate closed form likelihoods of sequences. This is in contrast to, say, energy based models which are generative models but that can only provide un-normalized likelihoods and are inefficient to sample from, or to GANs which can be sampled from but do not offer any way to estimate likelihoods. Specifically in the variant effect prediction space, most models considered are not proper generative models offering the ability to draw samples and calculate likelihoods efficiently. Thus, it is an important property of our model as we mention in that section.
>
> - Line 197, why the log-likelihood of the variant can be used as the fitness predictor? What's the intuition behind this? or can you provide the related reference?
>
> The log-likelihood or some un-normalized or approximate version thereof is used by every unsupervised variant effect predictor that we are aware of. The idea is that natural protein sequences must be evolutionarily fit and, therefore, by learning the generative distribution of these sequences we are capturing this density. The probability of a sequence variant is then reflective of fitness, because we assume that more fit sequences are more likely to be observed in nature (examples include refs [3, 4, 9, 10, 11, 13, 23] in main text).
>
> - Line 227, Comparison to baselines. There are many methods for fitness prediction in directed evolution, such as CLADE and CLADE2.0. Why not compare with them?
>
> These methods appear to be primarily supervised variant effect predictors, whereas we only consider the unsupervised variant effect prediction problem here and compare with other likelihood-based methods. These methods also only report results for four datasets, not for ProteinGym, a far more comprehensive collection of DMS datasets used here. In fact, for its unsupervised component, it appears that CLADE2.0 uses evolutionary scores from DeepSequence VAE, MSA Transformer, profileHMMs, and ESM-1v, all of which we already compare with, or compare with a better version of,and dramatically outperform.
>
> - In Table 1, What's the result of PoET if no ensemble?.
>
> We report results without ensembling in Appendix Table 4.

---

> > ### Comment · Reviewer_Uh6C · 2023-08-13
> >
> > I would like to thank the authors for the detailed rebuttal. After carefully reading the rebuttal, I think a partial of my concerns has been addressed. However, for a thorough evaluation, I would like to see more downstream task experiments beyond variant fitness.

---

### Official Review · Reviewer_sUvw · 2023-07-05

**Soundness:** 3 good
**Presentation:** 4 excellent
**Contribution:** 3 good
**Rating:** 8
**Confidence:** 3

**Summary:**

The authors of the paper present Protein Evolutionary Transformer (PoET), an autoregressive transformer-based model that is able to generate sets of related protein sequences, enabled by the proposed novel Transformer layer. This sequences-of-sequences generating method benefits from transfer learning, is able to handle novel indels, and does not need a multiple sequence alignment (MSA) as its input. The model can be conditioned on sequences of a protein family of interest for generation and scoring. The proposed method shows similar or superior performance compared to other models in fitness prediction tasks, while being faster without requiring retraining.

**Strengths:**

This paper is very well-written, with clear explanations, great examples and figures, and a comprehensive Related Work section. The experiments are explained well, they align with the claims made in the paper, and the results are discussed adequately. Moreover, there's an abundance of extra information in the appendix.

**Weaknesses:**

I enjoyed reading this paper, I do not have any major concerns. However, I did miss a "Limitations" section or something equivalent, either in the main text or in the supplementary. Moreover, I think some results (mainly those in the "Ablation" section 5.2) could be made stronger, for example by averaging performance over multiple runs and showing error bars, because the observed trends are not always that convincing. Finally, and this is more of a general issue, even though the results show improved/competitive performance, the average correlation values to experimental data is relatively low, i.e. around 0.5, which is still quite a weak correlation. The mismatch between density estimation and fitness prediction has been discussed before, for example in [1], and it might be worth discussing to some extent in this paper as well.

[1] Weinstein, Eli, et al. "Non-identifiability and the Blessings of Misspecification in Models of Molecular Fitness." Advances in Neural Information Processing Systems 35 (2022): 5484-5497.

**Questions:**

1. As mentioned in "Weaknesses": Limitations section missing.
2. As mentioned in "Weaknesses": perhaps include some discussion on the (mis)match between density estimation and fitness prediction.
3. As mentioned in "Weaknesses": if at all possible, it would be very informative to report averages over multiple runs, especially for the ablation results (Figure 3) but also for Table 1, to make the observations more convincing.

Minor comments:

4. The abstract states that PoET *outperforms* other models. However, from the number of boxed values in Table 1, this claim might be a bit too strong.
5. Do you report the dimensionality of embedding size $d$ somewhere? I might have missed it.
6. If I understand correctly, the relative positional encoding scheme would probably not be beneficial when there are big differences in sequence length amongst homologous proteins. In that case, there must be a better encoding possible (some MSA-like). Could you discuss this or perhaps mention it as a limitation?
7. The results on ensembling PoET with other models (Table 1 + Table 6 in the appendix) are interesting. Did you also try ensembling PoET with one or multiple other PoET model(s)?
8. Section 5.2.1:
    * In general, it could be considered "cheating" to monitor the correlation during training since this is essentially your test data. If it's just for these experiments then it's fine, I'm just checking if it's not something you used as a stopping criterion in general?
    * Context Length: apart from the earlier suggestion to do multiple runs here, it's also worth pointing out that the drawn conclusions depend quite strongly on when it was decided to stop training.
    * Model Size: the stopping criterion of "when the performance seemed to plateau" looks a bit arbitrary. If we compare the lines in the right graph in Figure 3, the blue line was cut off quite early while similar "plateaus" can be found in the purple and brown lines, which were allowed to train longer.
9. Missing references:
    * Figure 1a is never referenced.
    * Paper reference missing for RoPE?
    * Not all appendices are referenced in the main text (B, E, F, I, and O are missing).

**Limitations:**

Even though some limitations are touched upon in the main text, a thorough discussion of limitations is lacking.

---

> ### Author Rebuttal · Authors · 2023-08-10
>
> We thank the reviewer for their overwhelmingly positive response and comments. In answer to the reviewer’s questions:
>
> - As mentioned in "Weaknesses": Limitations section missing.
>
> We sought to discuss limitations throughout the paper as appropriate, but we will revise the paper to include an explicit “Limitations” section for the camera ready version.
>
> - As mentioned in "Weaknesses": perhaps include some discussion on the (mis)match between density estimation and fitness prediction.
>
> Yes, this is an extremely interesting point, which we mention briefly in Appendix D.2 Lines 399-412, but we will expand on this discussion in the revised manuscript.
>
> - As mentioned in "Weaknesses": if at all possible, it would be very informative to report averages over multiple runs, especially for the ablation results (Figure 3) but also for Table 1, to make the observations more convincing.
>
> We did not feel that multiple runs were necessary given the high cost of training these models and the fact that results are already averaged over a large number of datasets, which we think represents a more interesting source of variation, anyway, regarding expected performance on new variant effect prediction problems. We also do not have access to multi-run averages from other methods. Statistically significant claims relating to this source of variation are indicated with a box in Table 1.
>
> - The abstract states that PoET outperforms other models. However, from the number of boxed values in Table 1, this claim might be a bit too strong.
>
> We will adjust the text as suggested.
>
> - Do you report the dimensionality of embedding size d somewhere? I might have missed it.
>
> The dimensionality of the embedding is 1024. We report this in Appendix B.
>
> - If I understand correctly, the relative positional encoding scheme would probably not be beneficial when there are big differences in sequence length amongst homologous proteins. In that case, there must be a better encoding possible (some MSA-like). Could you discuss this or perhaps mention it as a limitation?
>
> The relative positional encoding scheme basically acts as a weak prior on the alignments between the sequences. We thought the same and have also trained a version of PoET without any positional information between sequences, which performs essentially identically to the version of the model we presented here. This suggests the model is able to learn the correct alignments and is not sensitive to this prior. We agree that there are other encodings which may be better, although, it is not-so-obvious how they can be applied correctly with an autoregressive decoder.
>
> - The results on ensembling PoET with other models (Table 1 + Table 6 in the appendix) are interesting. Did you also try ensembling PoET with one or multiple other PoET model(s)?
>
> We have not, but this would be interesting to try!
>
> - Section 5.2.1:
> - In general, it could be considered "cheating" to monitor the correlation during training since this is essentially your test data. If it's just for these experiments then it's fine, I'm just checking if it's not something you used as a stopping criterion in general?
>
> All hyperparameters, including training time and model size (Figure 3 caption) are tuned based on validation set performance only. This is consistent with other methods evaluated on ProteinGym.
>
> - Context Length: apart from the earlier suggestion to do multiple runs here, it's also worth pointing out that the drawn conclusions depend quite strongly on when it was decided to stop training.
> - Model Size: the stopping criterion of "when the performance seemed to plateau" looks a bit arbitrary. If we compare the lines in the right graph in Figure 3, the blue line was cut off quite early while similar "plateaus" can be found in the purple and brown lines, which were allowed to train longer.
>
> We agree that the stopping criterion can have a strong effect on results, and will note this as a limitation of the ablation. While we try to be as objective as possible, our ability to conduct ablations is ultimately constrained by compute resources.
>
> - Missing references:
> - Figure 1a is never referenced.
> - Paper reference missing for RoPE?
> - Not all appendices are referenced in the main text (B, E, F, I, and O are missing).
>
> We will correct these in the revised manuscript.

---

> > ### Comment · Reviewer_sUvw · 2023-08-16
> >
> > Thank you for addressing my questions and minor concerns point-by-point. I'll happily stick to my high score if indeed all promised changes are made. I understand that certain experiments are limited by time and compute requirements, but if the paper does get accepted, I would still suggest adding results for an ensemble of multiple PoET models for the camera-ready version for completeness.

---

### Official Review · Reviewer_jcwZ · 2023-07-10

**Soundness:** 2 fair
**Presentation:** 3 good
**Contribution:** 3 good
**Rating:** 7
**Confidence:** 4

**Summary:**

Current generative protein language models focus on generating individual protein sequences and are not specifically trained to generate sequences for an entire protein family. The paper introduces a new autoregressive generative model that tackles this limitation by generating Multiple Sequence Alignments (MSAs) for the entire family. The model utilizes a TieredTransformerDecoder to model the within-sequence interactions in an order-dependent manner, as well as the between-sequence interactions in an order-independent manner. The authors evaluate their models on protein variant fitness tasks by measuring the probability of variants conditioned on searched MSA of the parent sequence. The proposed method achieves large improvements on the ProteinGym benchmark. Importantly , the method addresses the problems faced by previous approaches, such as handling proteins with low-depth MSAs and accommodating indel mutations in addition to substitutions.

**Strengths:**

1. Protein variant fitness prediction tasks are very important for protein design. The approach, which achieves large improvements on this task, will definitely have large potential impact.
2. The related work section is well-documented, and it is relatively clear where the authors' contributions lie.
3. According to Table 1, the proposed method performs exceptionally well on proteins with low-depth MSAs, which supports the motivation behind generating MSA instead of individual sequences.
4. While the technique used in the paper is not entirely new, the idea of generating MSA instead of individual sequences is novel enough for an application-oriented paper.

**Weaknesses:**

1. The soundness of autoregressive decoding of MSA is questionable. Please see the questions below.
2. The authors claim their contribution as proposing a model for generating sequences for the entire family, yet few experiments are conducted to evaluate the generated MSAs.
3. Considering the significance and volume of this work, it would be disappointing if the code were not provided. The authors only promise to offer an accessible API, but it’s not sure whether they will provide the training code and trained models.

**Questions:**

Major points:
1. The author proposes to encode the MSA independently of the order between different sequences. But they still use an autoregressive decoding method that introduces order dependence in the output probabilities. For example, the probability equation $p(s_1)p(s_2|s_1)p(s_3|s_1,s_2)=p(s_1,s_2,s_3)=p(s_1,s_3,s_2)=p(s_1)p(s_3|s_1)p(s_2|s_1,s_3)$ may not hold when using the proposed autoregressive decoding. In the former case, generating tokens of $s_3$ is conditional on $s_1$ and $s_2$, whereas in the latter case, generating tokens of $s_3$ is conditional on only $s_1$.
Therefore, when measuring the fitness with $p(v,s_1,…,s_m)$, the results will differ using $p(v|s_1,…,s_m)p(s_1,…,s_m)$ and $p(s_{j+1},…,s_m|v,s_1,…,s_j)p(v|s_1,…,s_j)p(s_1,…,s_j)$. How is this problem addressed during inference? How is the order of MSA sequences determined during training?
2. A trivial baseline for generating MSA is to use an autoregressive language model, such as ProGen, conditioning the model on the parent sequence and continuing the generation with a start token afterward. If limited by the context window, we can only generate one sequence at each time. Although the model is not trained on MSA datasets, it is now well-known that language models trained with next-token prediction loss can excel at generating next sequences. How would this perform compared with the proposed approach?
3. The organization of the paper is problematic. It appears that the most significant contribution of this paper lies in the improvement of protein variant fitness prediction tasks. However, the title and introduction sections place more emphasis on the generation of MSAs, which lacks substantial support in their experiments (the main results and ablation study are all based on fitness prediction). I suggest the authors reconsider the introduction section to highlight the "proposal of a new approach for protein fitness prediction" rather than "proposal of a new approach for generating MSA."

Minor points:
1. In Sec. 5, it would be helpful to introduce the included baseline methods and provide the corresponding references before discussing the results.
2. In Table 1, the performance of TranceptEVE M on Indels is shown to be lower than that of TranceptEVE L. Could you provide an explanation for this observation?
3. Line 154 in Sec.3.1.2: while it may be well-known, it would be better to explain the abbreviation "RoPE" when it is first mentioned and include a reference at that point.

Overall, though with flaws, I think the paper makes a big contribution to the protein design community considering the amount of work in the paper. I’m willing to raise my score if the authors can address my concerns.


**Limitations:**

The authors should add a paragraph to discuss the potential limitations and negative societal impacts of their work.

---

> ### Author Rebuttal · Authors · 2023-08-10
>
> We thank the reviewer for their positive comments and excitement about the potential impact and performance of our method. In particular, we appreciate that the reviewer feels that our “...paper makes a big contribution to the protein design community considering the amount of work in the paper.” We respond to the reviewers specific comments and questions below.
>
> - The soundness of...
>
> To be clear, we do not decode MSAs. We decode sets of related protein sequences, which are unaligned.
>
> - The authors claim their contribution...
>
> The model is a generative model of whole protein families as sequences-of-sequences, but this full family generative task is not our primary application of interest. In the same way that GPT is a generative model of text, but is only used conditioned on prompts in practice, we are primarily interested in using our model as a conditional generative model, which is permitted by the structure of the model. The conditional generative capabilities enables both scoring of variants for fitness prediction and efficient generation of novel sequences via sampling. We have expanded our evaluation of the generative capabilities of the model (see global response), and will update the manuscript to include these in the main text.
>
> We have also explored unconditional generation of whole families with PoET, where we find that sampled families produce reasonable looking multiple sequence alignments, when aligned. We have also found that, interestingly, despite the lack of homology of these families to any natural proteins, conditioning AlphaFold2 on these multiple sequence alignments leads to feasible looking structure predictions with higher pDLLTs than are given to structure predictions using a single sequence from the sampled family alone. Unfortunately, we do not have enough space in the review supplement to include these results, and we don’t want to read too much into what this means, aside from the fact that PoET generates protein families that follow reasonable family-level statistical constraints. We will add some discussion of this to the manuscript.
>
> - Considering the significance...
>
> We will release code and the trained models with the camera ready version of the manuscript.
>
> - The author proposes...
>
> This is true, only the individual decoder layers are invariant to the order of the sequences (discussed at lines 131-132); the full model, composed of multiple such layers, is not. When considering variant fitness prediction, we only ever consider the conditional likelihood of the variant given the homologues, p(v | s1 … sn). In practice, we do not find ordering of the sequences to be an issue when considering the full joint likelihood p(s1 … sn), because the model is trained on random orderings of sequences, which requires it to learn to generate sequences given any ordering of the prior sequences. We will emphasize this fact in Section 3.2 for the revised manuscript.
>
> - A trivial baseline for generating ...
>
> ProGen is not designed to condition on sequences or to generate multiple sequences. It only conditions on control tags, which are generally annotations from Uniprot. Please see Review Supplement Table 1 in the global response to see how ProGen performs in terms of perplexity when conditioning on a parent sequence. We have also compared PoET with a baseline model that simply autoregressively generates the whole sequence-of-sequences in (Section 5.2.2 Figure 4, Appendix K Figure 10), which shows that general language models do not perform well at this task as they are unable to generalize beyond their training context length. In contrast, our specialized transformer layer allows PoET to generalize well to much longer context lengths.
>
> - The organization of the paper...
>
> We will revise the manuscript to better emphasize the variant effect prediction task. Also, please see the global response for additional analysis of the generative capabilities of PoET.
>
> - In Sec. 5, it ...
>
> We will revise the manuscript as suggested.
>
> - In Table 1, the performance of TranceptEVE M on Indels is shown to be lower than that of TranceptEVE L. Could you provide an explanation for this observation?
>
> These results are reported directly from the TranceptEVE paper. We also aren’t sure why TranceptEVE M performs better than L on indels, but it could be related to the mis-specification of the variant effect prediction task, or some other source we aren’t aware of. The number of datasets with indels is also significantly smaller than the substitution-only DMS datasets, which could contribute (the difference between M and L is not statistically significant, see lines 256-257).
>
> - Line 154 in Sec.3.1.2: ...
>
> We will revise the manuscript accordingly.
>
> - The authors should...
>
> We believe an explicit “Limitations” section is optional at NeurIPS this year and sought to discuss limitations throughout the manuscript where appropriate. We will collate these limitations and others raised in this discussion in a clear “Limitations” section in the revised manuscript.

---

> > ### Comment · Reviewer_jcwZ · 2023-08-16
> >
> > Thanks for the author's detailed response! The additional experiments performed during rebuttal clearly add value to the paper and my concerns are addressed by the authors' response. Therefore, I decide to raise my score to 7.

---

### Official Review · Reviewer_n5MZ · 2023-07-15

**Soundness:** 3 good
**Presentation:** 4 excellent
**Contribution:** 3 good
**Rating:** 7
**Confidence:** 5

**Summary:**

- This paper introduces PoET, a novel autoregressive transformer architecture that learns a distribution over protein families.
- More specifically, PoET takes as input a concatenated set of homologous sequences, for example retrieved with a MSA. The model architecture consists of a stack of several TieredTransformerDecoderLayers (the main innovation from a modeling standpoint), where each layer applies successively: 1) causal self-attention within each sequence 2) causal self-attention across sequences in the homologous set 3) a standard feedforward NN.
- From a practical standpoint, the model can be used at inference both to generate new proteins or to assess the effects of mutations on protein fitness
- The former (new sequence generation) is partially covered (one interesting yet limited example in appendix for chorismate mutase)
- The latter (fitness prediction performance) is the focus of experiments conducted in the paper. PoET achieves remarkable performance on the ProteinGym benchmarks using a 200M-param model

**Strengths:**

- The performance of the architecture on the zero-shot fitness prediction tasks from ProteinGym is quite remarkable for a model of this size
- The model architecture is simple and effective
- The different ablations to curate the optimal prompt engineering strategy are thorough and packed with many interesting insights

**Weaknesses:**

- The sequence generation abilities of the model are not really explored (besides one example for 1 protein family in appendix)
- This claim at the end of the introduction (lines 69-70) appears to be wrong: "improves prediction of variant effect in sequences with large numbers of mutations." Performance on deep mutants (5+ in Table S2) is lower than baselines (Tranception & TranceptEVE). Would suggest removing this sentence and adjust the second to last sentence of conclusion, as well as fixing the bolding in the appendix table.
- Some other claims are not properly substantiated. For example: "One advantage of PoET is that is able to not only score indels, but also generate sequences with indels" (lines 257-258). There does not seem any evidence in the paper that the quality of such "generated sequences with indels" would be any good, especially given the specifics of the position encoding chosen which relies on the assumption that "amino acids at similar absolute positions in homologous proteins are more likely to be drawn from the same distribution" (168-169). See questions for other claims where evidence seemed light.
- The performance seems to plateau as the number of parameters grow (the ~600M-param model performs on par with the ~200M-param version) and in terms of context length (12k better than 24k at inference) suggesting this architecture has perhaps already reached its limits

**Questions:**

- “Deep mutational scans and directed evolution experiments have been used to successfully design novel proteins [1, 2], but can be costly and difficult to implement,  which makes these experimental methods inapplicable for many proteins and functions of interest.” -- please elaborate on which proteins / functions DMS and directed evolutions are "inapplicable" (reference paper would be great to add here).
- “PoET is a fully autoregressive generative model, able to generate and score novel indels  in addition to substitutions, and does not depend on MSAs of the input family, removing problems caused by long insertions, gappy regions, and alignment errors.” -- isn't your method subject to these errors as well since you retrieve homologous sets at inference via a MSA?
- Any particular reason for using Diamond at training and then colabfold or jackhmmer at inference? Have you tried training on homologous sets retrieved with MSAs? (would make inference closer to training)
- Given your position encoding (which requires coordinate systems to be ~ equal across homologous sequences), isn't your method also not adapted to scoring sequences with long insertions/deletions?
- What are the limitations of your method? It seems very compelling for the majority of settings, except perhaps: 1) if one wants fast inference for good performance, GEMME is a better option 2) if one cares about deep mutants, Tranception / TranceptEVE (or ensembled with PoET) seem better 3) if one cares about disordered proteins / proteins with very few homologs (ie. less than 10), then it is not clear whether your model would handle well (since sets with fewer than 10 homologs were removed for training as per line 181). Would you agree with these limitations? How does the performance vary as the number of homologs goes to zero (note this would be a substantially different setting vs proteins in the "low MSA depth" bucket from ProteinGym)?
- Figure S1/S2: there is a reference to Prots2prot? Please also provide the performance at the DMS level for PoET alone
- Given the last point in the "weaknesses" box above, which avenue(s) do you see to further improve the performance of your model?

**Limitations:**

Not particularly discussed, would suggest adding a couple sentences about it, as per the questions above.

---

> ### Author Rebuttal · Authors · 2023-08-10
>
> We thank the reviewer for their positive comments and suggestions. In the manuscript, we strove to provide a comprehensive analysis of the prompt engineering aspects, which we hope will be relevant for using PoET and also for future work in retrieval-/homologue-augmented protein ML models. We’re glad the reviewer found this part interesting. We address the reviewer’s questions and concerns below. We will also expand the discussion of limitations to include points raised in this discussion.
>
> On PoET as a generative model, especially relating to the reviewer’s concerns about the quality of sampled indels: we focused our analysis on the variant effect prediction problem, because we view it as a good surrogate for generation in the sense that assigning high likelihood to high functioning variants means that samples or other decodings from the model, which are high likelihood, are also likely to be functional. In addition to our chorismate mutase example in the appendix, we have expanded our analysis of indel generation and will show some additional sampling examples in the revised manuscript. Please see our global response for more details. As suggested by other reviewers, we will also revise the introduction and other sections to better emphasize the variant effect prediction task.
>
> - This claim at the end of the introduction...
>
> We will revise the manuscript as suggested.
> - The performance seems ...
>
> This was one of the most surprising findings to us in this work. We also expected larger models and longer context lengths to perform better. We suspect this is related to the fundamental mis-specification of the variant effect prediction problem as density modeling, as recently discussed by [1] and in our Appendix (lines 399-412). We think this is also supported by the fact that holdout perplexity does improve at longer contexts lengths, but variant effect prediction does not.
>
> - “Deep mutational scans ...
>
> Fundamentally, deep mutational scanning can only be applied in settings where a high throughput functional assay is available, which usually means that variants can be produced in a single pot and the function of those variants can be read out via a selection and sequencing assay. Low throughput, expensive, and time consuming assays are not amenable to deep mutational scanning due to cost and time limitations. These references [2,3] discuss some of the challenges and considerations for developing functional assays. We will add these references to the text.
>
> - “PoET is a...
>
> We do not retain the alignment, passing only the unaligned sequences into PoET as the prompt. The challenge with operating on the MSA is not retrieval, but rather that large gaps disrupt the continuity of the sequences as viewed by axial transformer style models. We also do not assume a specific correct alignment.
> - Any particular reason...
>
> We used Diamond because it was the only tool available to perform the all versus all homology search in a reasonable amount of time. It is substantially faster than alternative homology search programs while attaining highly competitive performance.
> - Given your position...
>
> The relative positional encodings only provide a prior on the alignment. The model is able to, and does, learn how to properly correspond sequence elements within the sequences. In fact, in newer experiments, we removed the between-sequence relative positional encoding altogether and the model performs nearly identically, suggesting that this does not hinder model performance on long sequences and indels. We also note that our validation uniref50 clusters are extremely gappy and have high variability in sequence length within each cluster (see global response). Thus it directly measures our ability to model such indels.
> - What are the...
>
> We examine the generative performance of PoET, in terms of perplexity on heldout sequence clusters, using different numbers of homologous sequences, where we see good performance. In fact, PoET performs dramatically better than profileHMMs or other viable protein sequence models in this very few homologs regime (See Appendix Figure 10 for evaluation in terms of number of tokens in MSA and Review Supplement Table 2 in the global response for evaluation in terms of number of sequences), so we view this as a major advantage of PoET. Given our current understanding, we generally agree with your other comments. Clearly, GEMME is much faster than any neural network-based model, but we don’t think this is a significant limitation, especially given that we can efficiently sample from PoET as an alternative. Any kind of exhaustive evaluation of high order variants is likely to be intractable regardless of method chosen simply due to the huge size of those spaces.
> - Figure S1/S2:...
>
> Prots2prot was an old working name for PoET. We will correct this in the manuscript.
> - Given the last...
>
> Given that we think this is a problem related to misspecification of the unsupervised variant effect prediction problem, as well as the challenge in directly linking DMS fitness measurements with specific protein properties (often, many factors contribute to fitness and it may not be clearly associated with one property), this is a challenging question to answer. We think the ability to include additional conditional information in the prompt, such as structures, or specific property specifications could help, and would provide additional mechanisms for controlling the generative distribution and sequence generation.
>
> [1] Weinstein, Eli, et al. "Non-identifiability and the Blessings of Misspecification in Models of Molecular Fitness."
>
> [2] Fowler, D., Fields, S. Deep mutational scanning: a new style of protein science. Nat Methods 11, 801–807 (2014). https://doi.org/10.1038/nmeth.3027
>
> [3] Tiefenauer, L., & Demarche, S. (2012). Challenges in the Development of Functional Assays of Membrane Proteins. Materials, 5(11), 2205–2242. https://doi.org/10.3390/ma5112205

---

> > ### Comment · Reviewer_n5MZ · 2023-08-19
> > **Response to rebuttal**
> >
> > Dear authors,
> >
> > Thank you for the detailed responses and additional analyses provided during rebuttal. One point that was not addressed in your response is about the DMS-level performance of PoET alone (second-to-last question), ie., adding PoET alone to Fig S1/S2. This figure is quite insightful -- for instance, we see that ESM1v is performing fairly well across the board, but tanks on a handful of assays, bringing the overall average Spearman down. Since PoET is not shown there, I was thus curious about the variance of the performance across assays, relative to other baselines (eg., ESM1v being "high variance" and TranceptEVE "low variance"). Based on ablations in Tables S1-S3, performance seems relatively stable across settings, but perhaps you noticed something insightful at a more granular level?

---

> > > ### Author Response · Authors · 2023-08-21
> > > **Variance across DMS assays**
> > >
> > > We thank the reviewer for the clarification. We will add PoET alone to Fig S1/S2 in the revised manuscript.
> > >
> > > On a per dataset level, PoET performs very similarly to the ensemble of PoET + TranceptEVE L, with similar variance and slightly worse performance across the datasets. We did not notice any particular trends in this difference. Since we can't share images at this point of the discussion, we provide the table below which shows the 5th, 25th, 50th, 75th, and 95th percentiles across the substitution only DMS datasets for the models in Fig S1 and PoET alone. The statistics reflect the observation that ESM1v performance has higher variance; the interquartile range for ESM1v is higher than that of other models and it has markedly worse performance in the lower quantiles.
> > >
> > > |                      |   5th Percentile |   25th Percentile |   50th Percentile |   75th Percentile |   95th Percentile |
> > > |:---------------------|---------------:|----------------:|----------------:|----------------:|----------------:|
> > > | MSA Transformer      |        0.10825 |         0.34406 |         0.43761 |         0.51985 |         0.65331 |
> > > | ESM1v                |        0.01776 |         0.26636 |         0.45941 |         0.54331 |         0.65768 |
> > > | TranceptEVE L        |        0.18136 |         0.40970 |         0.48699 |         0.54856 |         0.67994 |
> > > | PoET                 |        0.18757 |         0.40236 |         0.50819 |         0.58754 |         0.69969 |
> > > | PoET + TranceptEVE L |        0.18879 |         0.42292 |         0.51518 |         0.59493 |         0.70788 |

---

> > > > ### Comment · Reviewer_n5MZ · 2023-08-21
> > > > **Final comments on rebuttal**
> > > >
> > > > Dear authors,
> > > > Thank you for the additional response. All my comments & questions have been addressed. Given the clarifications, additional experiments during rebuttal and minor corrections promised, I am now recommending acceptance more enthusiastically (increasing my score to 7).

---

### Author Rebuttal · Authors · 2023-08-10

Dear Reviewers,

We thank all of you again for your positive comments and constructive suggestions and questions. Here, we present additional analyses addressing some common themes in the reviews. This discussion is intended to be viewed with the figures in the attached review supplement PDF. We will include these results in the revised manuscript.

Additional analysis shows that PoET generates high quality, high diversity sequences.
- We have performed additional analysis of the sequences generated in the chorismate example and include an additional example on lysozyme. We have calculated the maximum sequence identity between our generated sequences and any natural protein and also folded the generated sequences using AlphaFold2 to show predicted structural conservation and pLDDT. PoET generates high diversity sequences that are predicted to be structurally similar, supporting the quality of sequences generated by PoET (Figure 1a for chorismate, Figure 1b for lysozyme).
- We have added an additional analysis of lysozyme sequences generated by PoET, including a comparison with lysozyme sequences generated by the fine-tuned ProGen model from Madani et al [0]. PoET generates higher diversity sequences than ProGen and these sequences are predicted to be more structurally conserved than sequences sampled from ProGen at similar levels of diversity. These structures are also predicted with high confidence by AF2, having high pDLLTs (>90% for almost all PoET generated sequences).

PoET was trained and evaluated on indel rich protein sequence clusters, has superior perplexities for these heldout sequences, and generates plausible indels in our chorismate mutase example.
- The uniref50 clusters in the validation set for perplexity evaluation are highly indel rich. An alignment for an exemplar cluster shows that there are large insertions and high diversity between sequences (Figure 2a). A histogram of the columns by percent gap shows that highly gappy columns are the most common (Figure 2b, left). Across all of the validation clusters (Figure 2b, right), this trend remains true, with alignments becoming more indel rich when more sequences (longer context lengths) are considered.
- PoET achieves low perplexities on these heldout sequences, showing that it is a good generative model of these indel rich families. It outperforms our baseline transformer, ProGen, and profileHMM, achieving better perplexities at each number of provided homologues (see Tables below and Figures 4 and 10 in the manuscript).
- In our chorismate mutase generative example, PoET generates sequences with novel indels and low sequence identity to natural chorismate mutases that are predicted to fold into the conserved chorismate structure with high pLDDT according to AlphaFold2 (Figure 3).

PoET achieves low perplexities on heldout Uniref50 clusters, outperforming ProGen and profileHMMs, even when conditioning on no or a very small number of homologs.
- We also evaluate heldout perplexity of ProGen continuing from a prompt sequence (as suggested by Reviewer jcwZ), where we find that PoET achieves better perplexity than ProGen without any conditioning, and when conditioned on the cluster seed sequences (Review Supplement Table 1). We note that our holdout set and ProGen’s holdout set are not the same, so sequences from our heldout Uniref50 clusters may occur in the ProGen training set and PoET still outperforms ProGen in this analysis.

| Model  | # Sequences Conditioned On | Mean Perplexity | Std. Dev |
|--------|----------------------------|-----------------|----------|
| ProGen |                          0 |       15.591333 | 5.524855 |
| ProGen |                          1 |       14.788089 | 4.022846 |
| PoET   |                          0 |       14.545277 | 2.941460 |
| PoET   |                          1 |       10.907637 | 2.817105 |
Review Supplement Table 1: Perplexity evaluation on our heldout Uniref50 clusters. ProGen and PoET are either conditioned on no homologs (0 sequences conditioned on) or the seed sequence from the cluster (1 sequence conditioned on) and perplexities are calculated on the remaining sequences. This evaluation only includes sequences with length <512, due to length limits for ProGen.

- PoET generalizes well from a very small number of homologs. Conditioning on only 5 homologs, PoET already achieves a perplexity of 7.9 on unseen family members, in contrast to 17.0 for the PSSM and 13.7 for the profileHMM (Review Supplement Table 2). This demonstrates that PoET is a significantly better generative modeling option for families with small numbers of homologs and generalizes remarkably well from small numbers of sequences.
| Model | # Sequences Conditioned On | Perplexity |
|-------|----------------------------|------------|
| PSSM  |                          1 |  18.541033 |
| PSSM  |                          5 |  16.982748 |
| PSSM  |                         10 |  15.912713 |
| HMM   |                          1 |  20.137338 |
| HMM   |                          5 |  13.716971 |
| HMM   |                         10 |  11.888966 |
| PoET  |                          1 |  10.173127 |
| PoET  |                          5 |   7.922363 |
| PoET  |                         10 |   7.252593 |
Review Supplement Table 2: Median perplexities of sequences from heldout Uniref50 clusters after conditioning on 1, 5, or 10 cluster members with a PSSM, HMM, or PoET. All sequences from all heldout clusters are included in this analysis.

We will incorporate these additional analyses into the revised manuscript to better support the generative modeling capabilities of PoET.

[0] Madani, A., Krause, B., Greene, E.R. et al. Large language models generate functional protein sequences across diverse families. Nat Biotechnol (2023). https://doi.org/10.1038/s41587-022-01618-2

---

> ### Author Response · Authors · 2023-08-13
> **Addendum: Sample Size for Statistics in Review Supplement Table 1**
>
> Dear Reviewers,
>
> The sample size to calculate the perplexities in Review Supplement Table 1 is 7228. Below, we have reproduced Review Supplement Table 1 with an additional column indicating the standard error of the mean perplexity. Using a paired t-test, the differences in mean perplexity between ProGen and PoET when both conditioned on 0 sequences and both conditioned on 1 sequence are statistically significant, with $p\ll1e-50$. We hope that this addendum helps to better contextualize the variance and statistical significance of the data in Review Supplement Table 1.
>
> | Model  | # Sequences Conditioned On | Mean Perplexity | Std. Dev | Std. Error |
> |--------|----------------------------|-----------------|----------|----------|
> | ProGen |                          0 |       15.591333 | 5.524855 | 0.06498 |
> | ProGen |                          1 |       14.788089 | 4.022846 | 0.04732 |
> | PoET   |                          0 |       14.545277 | 2.941460 | 0.03460 |
> | PoET   |                          1 |       10.907637 | 2.817105 | 0.03314 |
> Review Supplement Table 1: Perplexity evaluation on our heldout Uniref50 clusters. ProGen and PoET are either conditioned on no homologs (0 sequences conditioned on) or the seed sequence from the cluster (1 sequence conditioned on) and perplexities are calculated on the remaining sequences. This evaluation only includes sequences with length <512, due to length limits for ProGen.

---

> ### Author Response · Authors · 2023-08-18
> **Addendum 2: on sequence ordering**
>
> Dear reviewers,
>
> To evaluate the sensitivity of PoET to the order of sequences, we compare heldout sequence perplexities after conditioning on 10 sequences ordered either 1) randomly, following other experiments in our manuscript, 2) shortest-to-longest, or 3) longest-to-shortest. We also compare the full joint log likelihoods of these sequences of 10 sequences with the same orderings (Review Supplement Table 3).
>
>
> |                     | Perplexity of Next Sequence |                             |                                      | Joint Log Likelihood      |                             |                                      |
> |---------------------|-----------------------------|-----------------------------|--------------------------------------|---------------------------|-----------------------------|--------------------------------------|
> | Sequence Order      | Mean Perplexity             | Mean Difference from Random | Mean Relative Difference from Random | Mean Joint Log Likelihood | Mean Difference from Random | Mean Relative Difference from Random |
> | Random              | 7.44075                     | -                           | -                                    | -8099.74973               | -                           | -                                    |
> | Shortest-to-Longest | 7.44828                     | +0.00753 (0.07686)          | +0.00109 (0.01033)                   | -8123.18167               | -23.43149 (55.76313)        | -0.00335 (0.00838)                   |
> | Longest-to-Shortest | 7.44410                     | +0.00336  (0.07608)         | +0.00055 (0.01030)                   | -8091.89462               | +7.85511 (52.92428)         | +0.00105 (0.00746)                   |
>
> **Review Supplement Table 3**: Comparison of next sequence perplexities and joint log likelihoods, $\text{log } p(s_1, s_2, …, s_n)$, of sequences of 10 sequences with different orderings from heldout Uniref50 sequence clusters. We report mean perplexity and joint log likelihood as well as mean difference from random and relative difference from random for shortest-to-longest and longest-to-shortest orderings with the standard deviation across clusters in parenthesis. Relative difference is calculated as (a - r)/r where a is the ordered value and r is the value with random ordering.
>
>
>
> We find that sequence ordering has a little-to-no effect on next sequence perplexity and joint log likelihoods. Ordering sequences shortest-to-longest changes next sequence perplexity by only 0.00753 on average and has a relative difference of <0.5% from random for both next sequence perplexity and joint log likelihood. Contrasting this with the next sequence perplexities of HMMs and PSSMs conditioned on 10 sequences from Review Supplement Table 2, we can see that PoET performs significantly better regardless of sequence ordering.

---

> > ### Comment · Reviewer_UQRE · 2023-08-18
> >
> > Thanks for authors quick response and experiments to clarify the sensitive analysis of sequence ordering. The reported experiments indicated that the performance of PoET is not sensitive to the input order. This is quite helpful in practical application. I suggest adding this sensitive analysis results in the revision version. And I increase my score to 5 as positive rating (Borderline accept).

---

> > > ### Author Response · Authors · 2023-08-18
> > >
> > > Likewise, thanks for the quick response! If you could, please also edit your score in your review to reflect this update.

---

### Decision · Program_Chairs · 2023-09-21

**Decision:**

Accept (poster)

**Comment:**

This submission draws unanimous praise from the reviewers for its novelty and clarity. Here are the key takeaways based on the collective feedback:

* One of the standout features is strong performance achieved by the proposed model architecture on the zero-shot fitness prediction tasks from ProteinGym. Notably, this was accomplished using a simple yet powerful model architecture. The authors present various ablations, shedding light on the ideal prompt engineering strategy.

* Novel Contributions: While building upon existing techniques, the paper introduces the inventive approach of generating MSA over individual sequences. This novelty accentuates the paper's credentials, especially for an application-driven submission. The introduction of a new Transformer layer that captures both order-dependence within sequences and order-independence between sequences. The results presented, particularly the superior performance of the proposed PoET over existing models on deep mutational scanning datasets in ProteinGym

Given the positive consensus among reviewers, the submission is suitable for acceptance at NeurIPS.